# Robo signalling controls pancreatic progenitor identity by regulating Tead transcription factors

Sophie Escot[1], David Willnow[1,2], Heike Naumann[1], Silvia Di Francescantonio[1] & Francesca M. Spagnoli[1,2,3]

A complex interplay of intrinsic factors and extrinsic signalling pathways controls both cell lineage commitment and maintenance of cell identity. Loss of defined cellular states is the cause of many different cancers, including pancreatic cancer. Recent findings suggest a clinical role for the conserved SLIT/ROBO signalling pathway in pancreatic cancer. However, whilst this pathway has been extensively studied in many processes, a role for *Slit* and *Robo* genes in pancreas cell identity and plasticity has not been established yet. Here, we identify Slit/Robo signalling as a key regulator of pancreatic progenitor identity. We find that *Robo1* and *Robo2* are required for preserving pancreatic cell identity shortly after fate induction and, subsequently, for expansion of the pancreatic progenitor pool in the mouse. Furthermore, we show that Robo receptors control the expression of Tead transcription factors as well as its downstream transcriptional activity. Our work identifies an interplay between Slit/Robo pathway and Tead intrinsic regulators, functioning as gatekeeper of pancreatic cell identity.

[1] Lab. of Molecular and Cellular Basis of Embryonic Development, Max-Delbrueck Center for Molecular Medicine, Robert-Roessle Strasse 10, Berlin 13125, Germany. [2] Berlin Institute of Health (BIH), Berlin 10178, Germany. [3] Centre for Stem Cell and Regenerative Medicine, King's College London, Great Maze Pond, London SE1 9RT, UK. Correspondence and requests for materials should be addressed to F.M.S. (email: francesca.spagnoli@kcl.ac.uk)

The Roundabout (Robo) receptors and their secreted Slit glycoprotein ligands[1] were originally identified as important axon guidance molecules, serving as a repulsive cue to allow precise axon path finding and neuronal migration during development[2,3]. In recent years, the functional repertoire of Slits and Robo has expanded tremendously, also outside of the nervous system[4], in the development of other tissues, such as the lung, kidney and mammary gland[5–10]. In particular, Slit/Robo signalling has been linked to a variety of fundamental processes, including cell adhesion, proliferation, survival and fate specification, depending upon the tissue context[1,11–14]. Importantly, a number of studies have linked these guidance molecules to pancreatic cancer and, in particular, to pancreatic ductal adenocarcinoma (PDAC), a devastating malignancy with an extremely poor prognosis[15–17]. Studies of pancreatic cancer clinical cohort identified recurrent mutations and copy-number variations in SLIT2, ROBO1 and ROBO2, suggesting a role in PDAC initiation and progression[15]. Recent observations suggested a role for local Slit secretion in the survival and function of pancreatic beta-cells in the adult pancreas[14], while Robo receptors are required in the beta-cells for endocrine cell sorting and mature islet architecture[18]. However, it is not known whether Slit/Robo signalling pathway functions in the establishment and maintenance of pancreatic cell identity and differentiation. Here, we report that Robo receptors act as key regulators of pancreatic progenitor cell identity. Here we find that *Robo1* and *Robo2* are expressed in pancreatic progenitors from the time of their fate specification, while the *Slit3* ligand is expressed in the surrounding mesenchyme. *Robo* inactivation in the mouse results in the loss of pancreatic cell identity and reduced pool of pancreatic progenitors by impinging on the TEAD transcription factors. Our findings reveal a role for Slit/Robo in pancreatic progenitors, expanding the vast array of biological functions already attributed to this conserved pathway.

## Results

**Loss of Robo receptors results in reduced pancreas organ size.** By RNA-sequencing (RNA-Seq) analysis on cells isolated from mouse foregut endoderm and embryonic pancreas between embryonic stage (E) 8.5 to E10.5[19], we discovered a spatially distinct expression of *Robo1* and *Robo2* in early pancreatic progenitors. The embryonic pancreas arises from the endoderm as two distinct rudiments, the ventral and dorsal pancreatic buds, which ultimately fuse together to form the adult organ[20,21]. In line with the RNA-Seq data[19], at E10.5 we found that *Robo1* and *Robo2* are abundantly expressed in ventral pancreatic progenitors and very low or absent in dorsal pancreas and liver (Fig. 1a, b). *Robo2* exhibited high expression levels already in the ventral foregut endoderm at E8.5, before pancreatic fate specification has occurred[21] (Supplementary Fig. 1a). Among the *Slit* ligand genes, *Slit3* was found enriched in the mesenchyme surrounding the ventral pancreatic epithelium at E10.5, in a complementary pattern with *Robo2* (Fig. 1a and Supplementary Fig. 1b). Subsequently, both *Robo1* and *Robo2* continued to be expressed in the pancreas throughout embryonic development (Fig. 1b). In humans, both ROBO1 and ROBO2 receptors were present in adult pancreatic islets (Supplementary Fig. 1d), like in adult mouse pancreas[14].

To study whether the Slit/Robo pathway controls pancreas organogenesis, we started by examining all stages of pancreatic development in *Robo1* and *Robo2* double mutant mice[3,7] (hereafter referred to as Robo1/2 KO). Non-transgenic littermates served as negative controls. Overall, we found that Robo1/2 knockout (KO) embryos display a severe organ size reduction at birth, which strongly affected the head of the pancreas, a

derivative of the ventral pancreas (Fig. 1c, d and Supplementary Fig. 2a). Pancreas formation was not affected in single Robo1 KO or Robo2 KO mutant embryos, suggesting functional redundancy in this context (Supplementary Fig. 2b). Further analysis of the mouse mutant phenotype showed that all pancreatic cell types are present in Robo1/2 KO pancreata and appropriate ratios between cells of different types are maintained, which rule out major defects in differentiation (Fig. 1e).

**Robo receptors are required for preserving pancreatic identity.** The pancreas final organ size is predetermined during embryonic development, being primarily dependent on the number of progenitor cells[22]. We therefore tested if the *Robo* genes control pancreatic embryonic size. In the absence of *Robo1* and *Robo2*, pancreatic progenitors were specified, as judged by the presence of Pdx1-positive cells at E9.5 (Fig. 2a). However, the volume of Robo1/2 KO ventral pancreatic bud was strongly reduced already at E9.5, while liver and dorsal pancreas rudiments were not affected (Figs. 2a, 3d and Supplementary Fig. 3). As expected, control ventral pancreatic progenitors displayed high levels of Pdx1 and low Prox1 transcription factors[19,20] (Fig. 2a, b). By contrast, single-cell examination of Robo1/2 KO embryonic ventral pancreas revealed the presence of cells that are negative or faintly positive for pancreatic transcription factors, including Pdx1 and Sox9, but high for Prox1, like hepatoblasts at this developmental stage[23] (Fig. 2a, b). Importantly, Prox1-positive cells intermingled with Pdx1/Prox1-double positive cells throughout the Robo1/2 KO ventral organ bud, but presented distinct morphological features, being mostly organised in small clusters that appeared segregated from the surrounding cells and displaying low E-cadherin membrane staining (Fig. 2a, b). When single-cell examination analysis of Prox1 and Pdx1 fluorescence intensities was performed at E8.5, no obvious difference was apparent between control and Robo1/2 KO ventral foregut endoderm (Supplementary Fig. 3a). This result suggested that *Pdx1* is normally induced in ventral pancreatic progenitors at the right developmental time, but in the absence of Robo signalling its expression cannot be maintained throughout the epithelium.

Next, we analysed cell death and proliferation in Robo1/2 KO embryonic pancreata. No cleaved-Caspase3 (Cas3)-positive cells were detected in control pancreata at E9.5, while in the absence of both *Robo1* and *Robo2* a significant number of Cas3-positive cells were found specifically in the ventral pancreatic bud (Fig. 3a, b and Supplementary Fig. 3b,c), suggesting ongoing apoptosis. Approximately 24 h later, starting from E10.5 we found proliferation defects in the ventral pancreas of mutant embryos, as judged by the reduced number of phospho-histone H3 (pHH3)-positive cells (Fig. 3e, f). At this embryonic stage, the reduction of the ventral pancreatic buds became more pronounced, affecting both Pdx1-positive and Sox17-positive buds but not yet the dorsal pancreas (Fig. 3c, d and Supplementary Fig. 3d). This is consistent with the higher levels of expression of *Robo* genes in ventral pancreatic progenitors compared to the dorsal ones at early stage (Fig. 1). Later in development, the proliferation defects were maintained in the ventral bud and also became detectable in the dorsal one (Supplementary Fig. 3f), possibly contributing to the overall organ size reduction (Fig. 1).

Because the pancreas shares a common origin with the liver and gall bladder[23–25], we analysed whether these organs are affected in Robo1/2 KO. We found that the liver size and tissue architecture were normal in Robo1/2 KO embryos, whereas the gall bladder was reduced in size, suggesting a specific requirement of the Robo signalling in the pancreato-biliary tract (Supplementary Fig. 4). Taken together, these results suggest that Robo receptors play a role in preserving

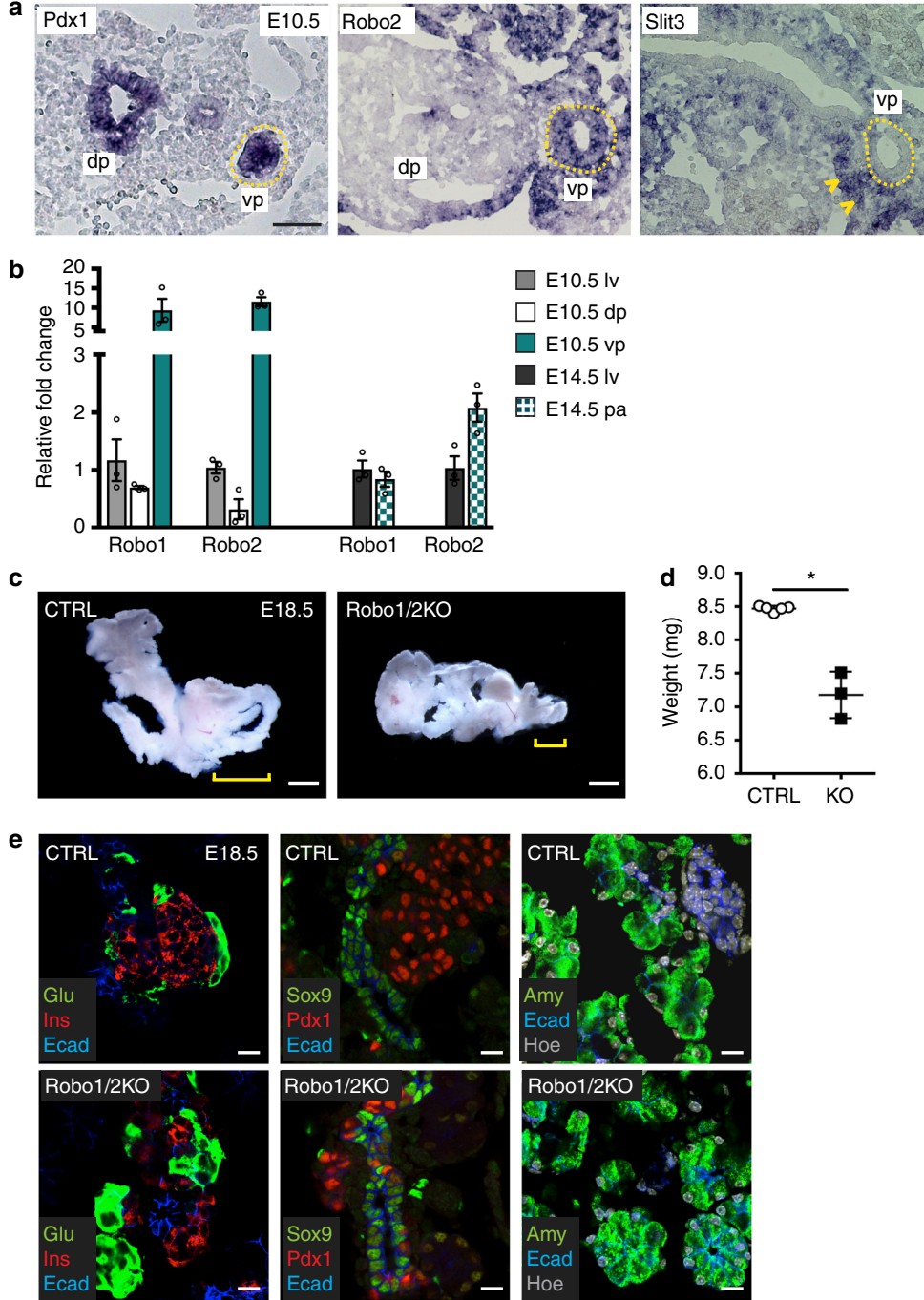

**Fig. 1** *Robo* genes control pancreas organ growth. **a** In situ hybridisation analysis of *Pdx1*, *Robo2* and *Slit3* on E10.5 mouse cryosections. Yellow dotted lines demarcate ventral pancreatic epithelium; yellow arrowheads indicate *Slit3* expression in the surrounding pancreatic mesenchyme. dp dorsal pancreatic bud, vp ventral pancreatic bud. Scale bar, 100 μm. **b** Quantitative RT-PCR (RT-qPCR) analysis of *Robo1* and *Robo2* expression in mouse liver and pancreas at E10.5 (left) and E14.5 (right). Data represent mean ± s.e.m.; *n* = 3. **c** Pancreas gross morphology at E18.5. Yellow brackets indicate the head of the pancreas. Scale bars, 1 mm. **d** Quantification of pancreas weight in control (CTRL; *n* = 5) and Robo1/2 KO (KO; *n* = 3) E18.5 mouse embryos. Error bars represent ± s.e.m. *Two-tailed Student's *t*-test *P* < 0.05. **e** Immunofluorescence (IF) analysis of Glucagon (Glu), Insulin (Ins), E-cadherin (Ecad), Amylase (Amy), Sox9 and Pdx1 on CTRL and Robo1/2 KO E18.5 mouse pancreatic cryosections. Scale bars, 20 μm

pancreatic identity and expansion of the number of ventral pancreatic progenitors.

**Robo inactivation leads to an endoderm metastable cell state.** The cells, which were Pdx1 low or negative in Robo1/2 KO ventral pancreata, displayed high Prox1 levels, comparable to

hepatoblasts, as well as reduced E-cadherin at the membrane (Fig. 2), resembling cells leaving the plane of the epithelium. Given these features, we then asked whether in the absence of *Robo1* and *Robo2* these cells had undergone a fate switch and acquired a liver fate, which is the closest embryonic fate[23,25]. First, we observed that the cells, which were Prox1 positive and Pdx1 negative in Robo1/2 KO ventral pancreas, were also positive

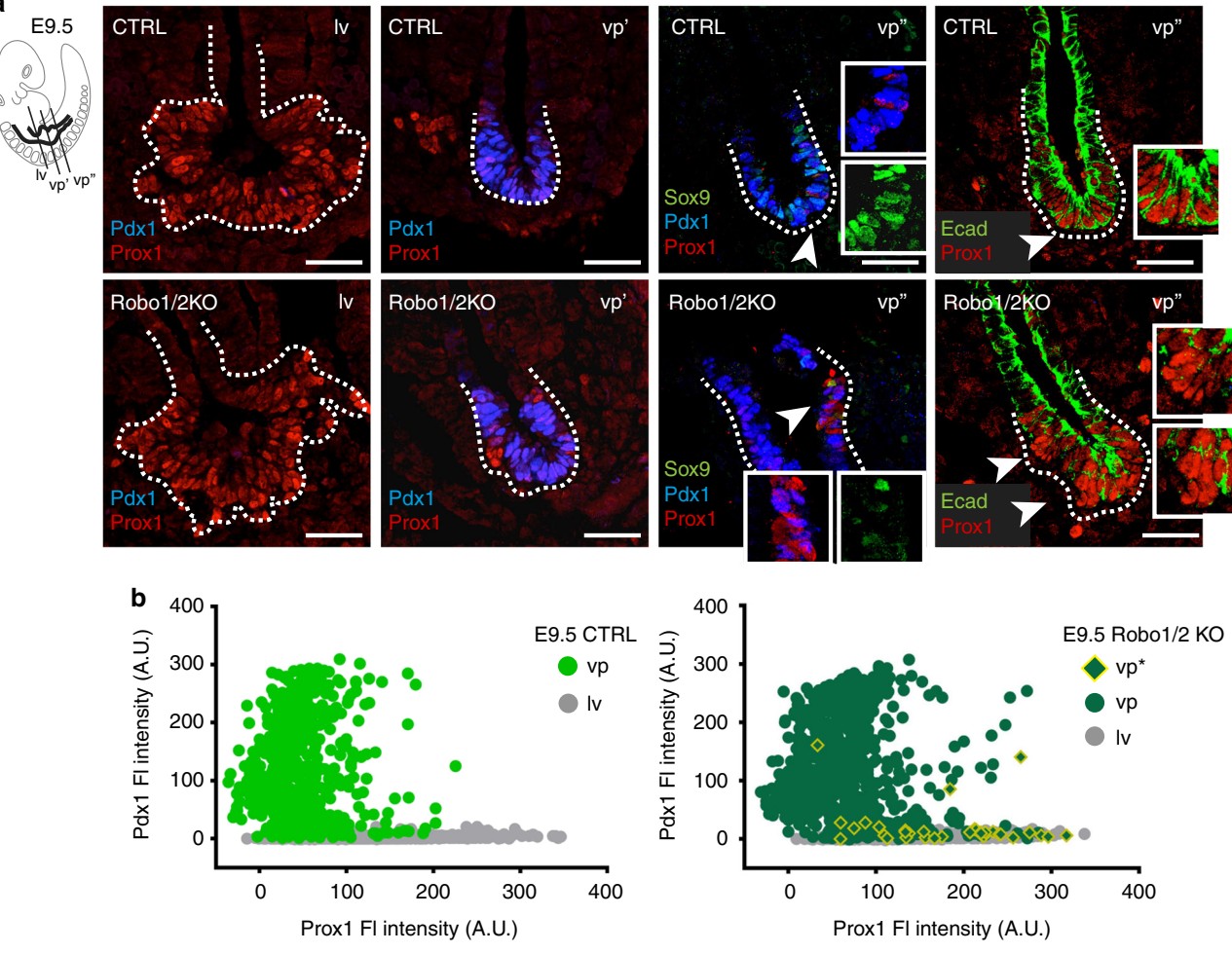

**Fig. 2** *Robo* genes preserve pancreatic progenitor identity. **a** Representative IF images of E9.5 CTRL and Robo1/2 KO cryosections stained for Pdx1, Prox1, Sox9 and E-cadherin (Ecad). Consecutive liver (lv) and ventral pancreas (vp', vp") sections are shown. Liver (lv) cells are Prox1-positive ($^+$), pancreatic progenitors (vp) are Prox1$^+$/Pdx1$^+$. Arrowheads indicate cells shown in white insets. Scale bars, 50 μm. **b** Single-cell measurement of fluorescence intensity of Prox1 and Pdx1 in liver buds (grey circle) and ventral pancreas of E9.5 CTRL (light green circle) and Robo1/2 KO (dark green circle) embryos. Diamond shapes indicate Robo1/2 KO ventral pancreatic cells that appear segregated from the surrounding epithelium and overlay with lv cells (grey). Fluorescence intensity was measured with Fiji and values corrected by linear normalisation within each embryo. FI fluorescence intensity, AU arbitrary units; $n = 3$ per genotype

for liver markers, such as Albumin and Alpha-fetoprotein[23] (Fig. 4a and Supplementary Fig. 3g). Furthermore, these changes were associated with actin cytoskeleton reorganisation and basal membrane disruption. Specifically, pancreatic progenitors showed prominent accumulation of F-actin at the apical surface, whereas Robo1/2 KO cells with high Prox1 displayed changes in F-actin distribution, showing also basal and lateral Phalloidin staining, which was used for staining F-actin (Fig. 4b). Concomitantly, we found breakdown of the laminin-enriched basal membrane, which normally surrounds the ventral pancreatic bud at this embryonic stage, and induction of high levels of *Hhex* in the ventral foregut of Robo1/2 KO embryos compared to controls (Fig. 4c). Both *Hhex* and *Prox1* transcription factors are known to be essential for delamination and migration of early hepatoblasts from the hepatic diverticulum[26,27]; thus, the morphological changes induced upon the loss of Robo receptors are reminiscent of early liver morphogenesis and might imply delamination and migratory behaviour of mutant pancreatic cell clusters.

Next, we traced the behaviour of pancreatic progenitor cells using lineage tracing analysis in *Pdx1*-Cre;R26R-H2B-GFP; Robo1/2 KO mouse embryos[3,7,28,29]. At E10.5 and E12.5, we observed an elevated number of ectopic green fluorescent protein

(GFP)-positive cells in the liver of Robo1/2 KO embryos (Fig. 5a, b). Quantitation of the number of GFP-positive cells in the liver bud showed that mutant embryos have about eight-fold higher number of GFP-positive cells relative to controls (Fig. 5b, c). Notably, the GFP-positive cells found in the liver were negative for Pdx1, but expressed hepatocyte markers, such as Prox1 and Albumin, as well as biliary epithelial cell markers, such as CK19, throughout development until birth (Fig. 5d, e). Taken together, these results suggested that in the absence of *Robo1* and *Robo2* a subset of pancreatic progenitors lose their original identity and switch to a liver cell fate.

The overall patterning of the primitive gut and surrounding organ rudiments was not affected in the absence of both Robo1 and 2 receptors (Supplementary Fig. 5a). Nevertheless, given the expression of *Robo* genes also in the surrounding microenvironment (Fig. 1a), we sought to directly assess the cell-autonomous activity of Robo receptors in pancreatic cells. To this aim, we conditionally inactivated *Robo2* gene in the ventral pancreatic epithelium using a loxP-flanked *Robo2* allele[8] and the *Pdx1*-Cre transgenic strain[29] (Fig. 5c and Supplementary Fig. 2d) in a *Robo1*-deficient background[3,7], hereafter referred to as Robo$^{PaΔ}$. Robo$^{PaΔ}$ embryos showed pancreatic defects similar to those

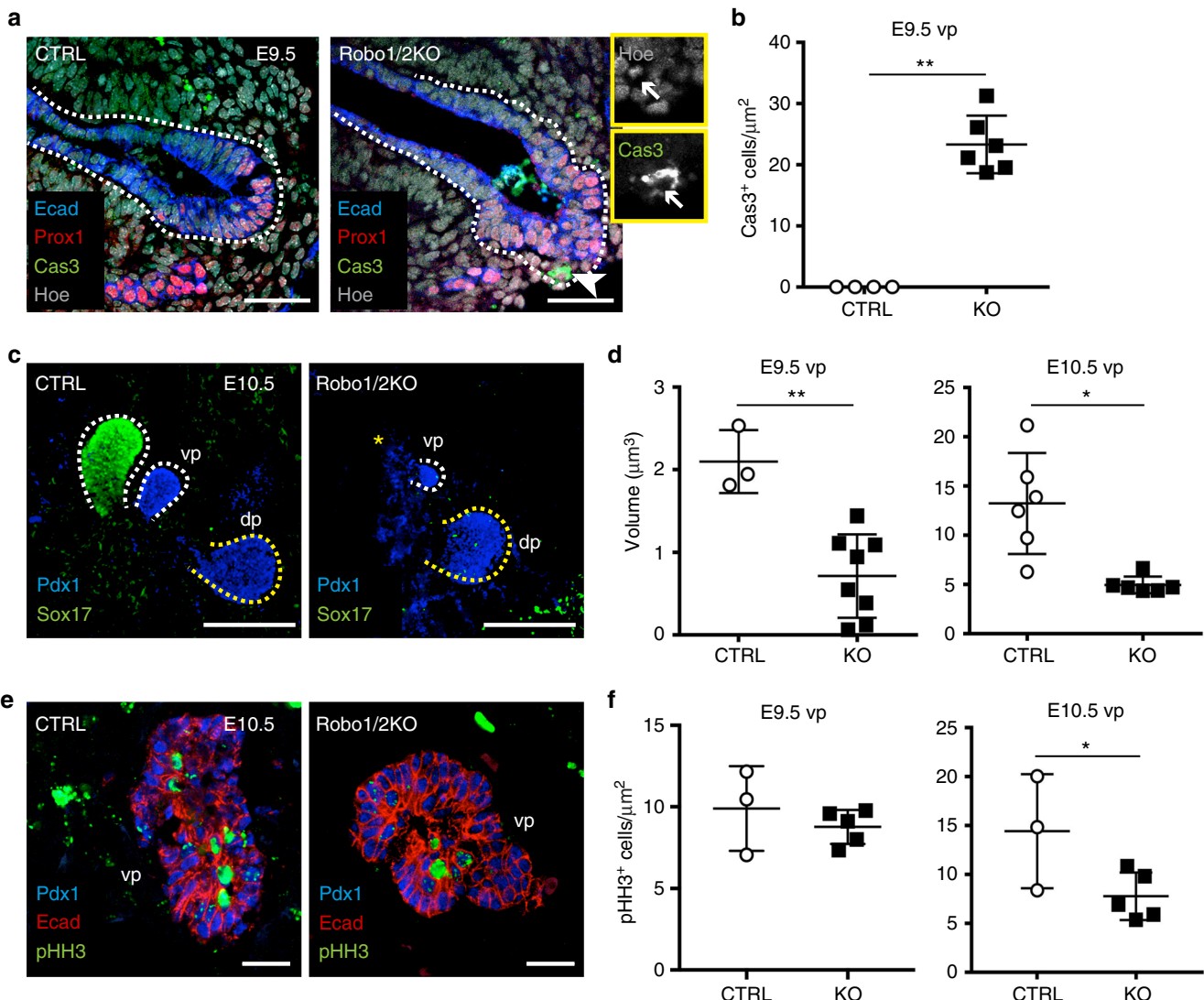

**Fig. 3** Robo receptors are required for cell survival and proliferation of ventral pancreatic progenitors. **a** Representative IF images of E9.5 CTRL and Robo1/2 KO cryosections stained for Ecad, Prox1 and cleaved-Caspase3 (Cas3). Hoechst (Hoe) nuclear counterstain. Arrowhead indicates apoptotic nucleus; in yellow insets are shown Hoechst and Cas3 single channels. Scale bars, 50 µm. **b** Quantification of Cas3$^+$ cells in ventral pancreata of E9.5 CTRL ($n = 4$) and Robo1/2 KO ($n = 6$) embryos. Number of Cas3$^+$ cells was normalised to the sum of the pancreatic Ecad$^+$ epithelial areas (µm$^2$). **c** Representative confocal maximum intensity z-projections of whole-mount IF for Pdx1 and Sox17$\alpha$ on E10.5 CTRL and Robo1/2 KO embryos. White dashed line marks ventral pancreatic bud; yellow dashed line marks dorsal pancreatic buds. Absence of Sox17$^+$ ventral pancreatic bud in KO embryo is indicated by yellow asterisk (*). Scale bars, 50 µm. **d** Quantification of E9.5 and E10.5 ventral pancreas volume from confocal images of CTRL and Robo1/2 KO embryos. **e** Representative IF images of E10.5 CTRL and Robo1/2 KO cryosections stained for Pdx1, phospho-Histone H3 (pHH3) and Ecad. Scale bars, 50 µm. dp dorsal pancreas, vp ventral pancreas. **f** Quantification of proliferation in ventral pancreatic buds of E9.5 and E10.5 CTRL and Robo1/2 KO cryosections. Number of pHH3$^+$ cells was normalised to the sum of the Ecad$^+$ epithelium area (µm$^2$). Error bars represent ±s.e.m. Two-tailed Student's t-test *$P < 0.05$, **$P < 0.01$

observed in Robo1/2 global loss-of-function mutation at birth as well as at earlier embryonic stages, albeit less severe (Fig. 6a–d and Supplementary Fig. 6a). This is likely due to slightly later inactivation of Robo2 within the endoderm compared to global Robo1/2 KO.

Next, we used the mouse embryonic stem cell (mESC) ex vivo system to model pancreatic fate induction using a previously published step-wise differentiation protocol[30,31]. First, reverse transcription–quantitative PCR (RT-qPCR) analysis showed induction of *Robo* genes in definitive endoderm (DE) and pancreatic endoderm (PE) cells when compared with undifferentiated cells, like in vivo in mouse embryos (Supplementary Fig. 1c). This was mirrored by the concomitant expression of *Slit* ligand genes in ES cultures undergoing PE differentiation

(Supplementary Fig. 1c). Therefore, to block the Slit/Robo signalling pathway ex vivo, we used the soluble Robo2-Fc chimeric receptor, which is commonly used as Slit ligands trap[32,33]. Following the treatment of differentiating mESCs with recombinant Robo2-Fc chimera, we found that pancreatic differentiation is impaired, as judged by the decrease in expression levels of *Sox17α*, *Foxa2*, *Hnf1b*, *Pdx1* and *Nkx6.1* (Fig. 6e). Particularly affected were marker genes, which are hallmarks of ventral pancreas endoderm, such as *Sox17α* and *Hnf1b*[19–21], whereas sustained expression of transcription factors in common between hepatic and pancreatic progenitors, such as *Prox1* and *Hex*, was maintained in the presence of Robo2-Fc (Fig. 6e). Consistent with the RT-qPCR results, immunofluorescence (IF) analysis revealed a significant reduction of the number

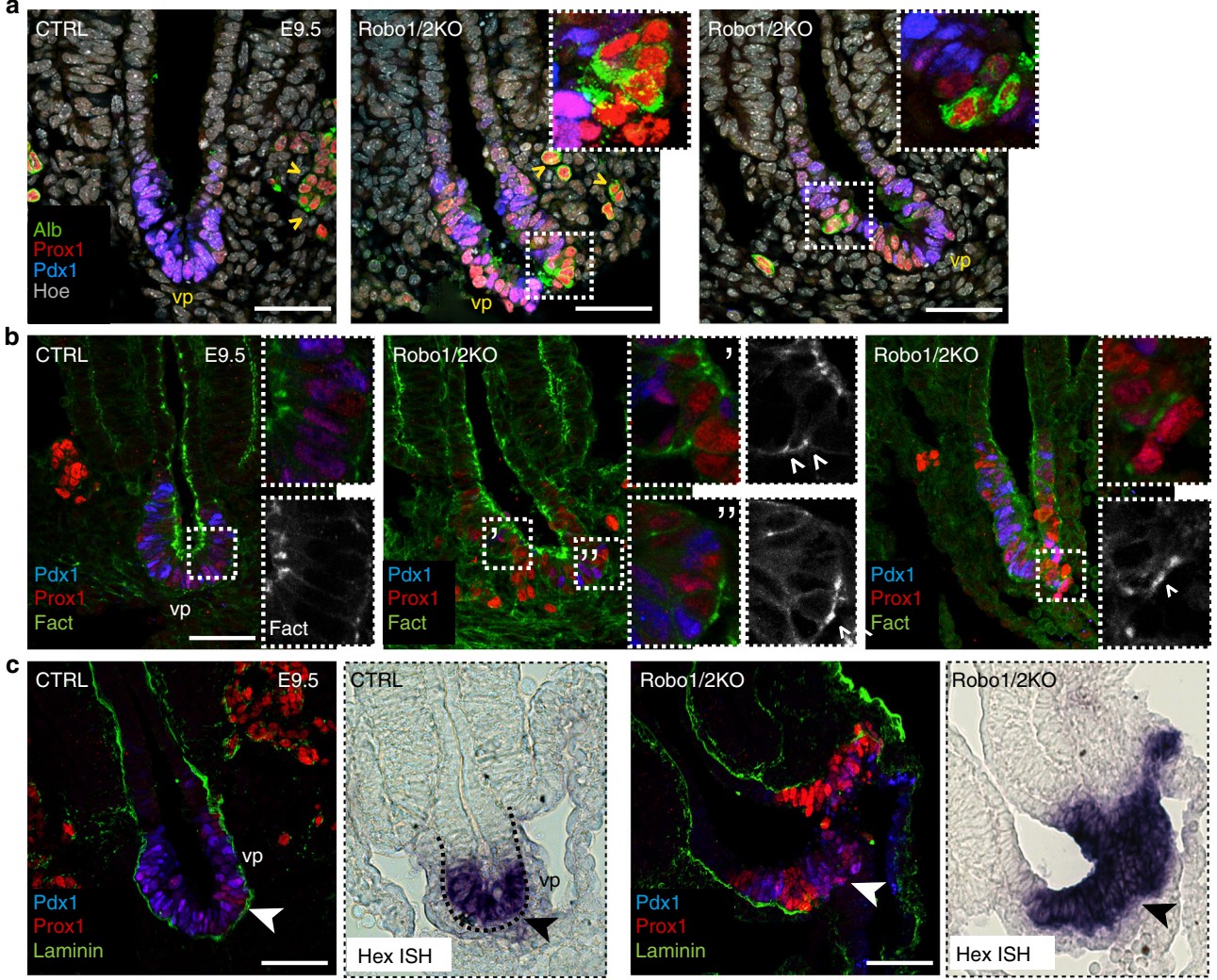

**Fig. 4** In the absence of *Robo1* and *Robo2* pancreatic cells acquire an hepatoblast identity. **a** Representative IF images of E9.5 CTRL and Robo1/2 KO cryosections stained for Albumin (Alb), Pdx1 and Prox1. Hoechst (Hoe) was used as nuclear counterstain. Arrowheads indicate Prox1+/Alb+ liver progenitors in hepatic cords. In dashed box, Pdx1-, Prox1+ and Alb+ cells within the ventral pancreatic epithelium (vp) of KO embryos. Insets show boxed area at higher magnification without Hoe channel. Scale bars, 50 μm. **b** Representative IF images of E9.5 CTRL and Robo1/2 KO cryosections stained for Phalloidin (F-actin), Pdx1 and Prox1. Insets show boxed area at higher magnification and Phalloidin (F-act) channel alone. Note the accumulation of F-actin at the lateral or basal surface of the mutant cells (arrowheads). **c** Representative IF images of E9.5 CTRL and Robo1/2 KO cryosections stained for Laminin, Pdx1 and Prox1. In boxed area, consecutive sections stained using antisense *Hhex* ISH probe. Arrowheads indicate laminin-enriched basal membrane surrounding the ventral pancreatic bud. Scale bars, 50 μm

of Pdx1-positive cells after inhibition of Robo/Slit signalling when compared to untreated PE cultures (Fig. 6f and Supplementary Fig. 6b,c).

To further distinguish between cell-autonomous and non-cell-autonomous contribution to the pancreatic phenotype, we analysed pancreatic vascularisation and innervation[34,35] in the absence of *Robo1* and *Robo2* (Supplementary Fig. 5b). At E10.5, E12.5 and E18.5, endothelial cells and TuJ1-positive neurons were present around and within Robo1/2 KO pancreas, comparable to control embryos. Taken together, these findings suggested that *Robo2* is specifically required within the pancreatic epithelium to regulate cell identity and survival.

**Robo signalling interacts with the YAP/Tead pathway.** To start dissecting the molecular network downstream of Robo signalling in pancreas progenitors, we analysed the expression of key target genes for signalling pathways that are known to be involved in early pancreas formation, including Wnt, YAP/Tead, TGFβ, BMP and Notch[20,21]. Slit/Robo signalling has been shown to interact with the Wnt/β-catenin signalling in different types of tumour cells[1,13], and we found here that Robo1/2 KO pancreatic cells exhibit a reduced level of *Axin2* expression but *Lef1* appeared unchanged at E12.5 (Fig. 7a). Additionally, the expression of the transcription factor *Tead2* and other well-known target genes of the YAP pathway, such as *Ctgf* and *Axl*[36,37], were reduced in the Robo1/2 KO pancreas, whereas target genes tested for the other pathways were unchanged (Fig. 7a). The Hippo/YAP pathway is a conserved major player in organ size control and cancer[38–41], with the Tead family of transcription factors being major mediators of the YAP biological outcome[36]. Recent findings in humans characterised the TEAD factors as integral components of the enhancer network in pancreatic progenitors[42]. TEAD and its coactivator YAP have been shown to activate key pancreatic transcription factors and promote the expansion of pancreatic progenitors[42]. Additionally, all *Tead* genes, especially abundant

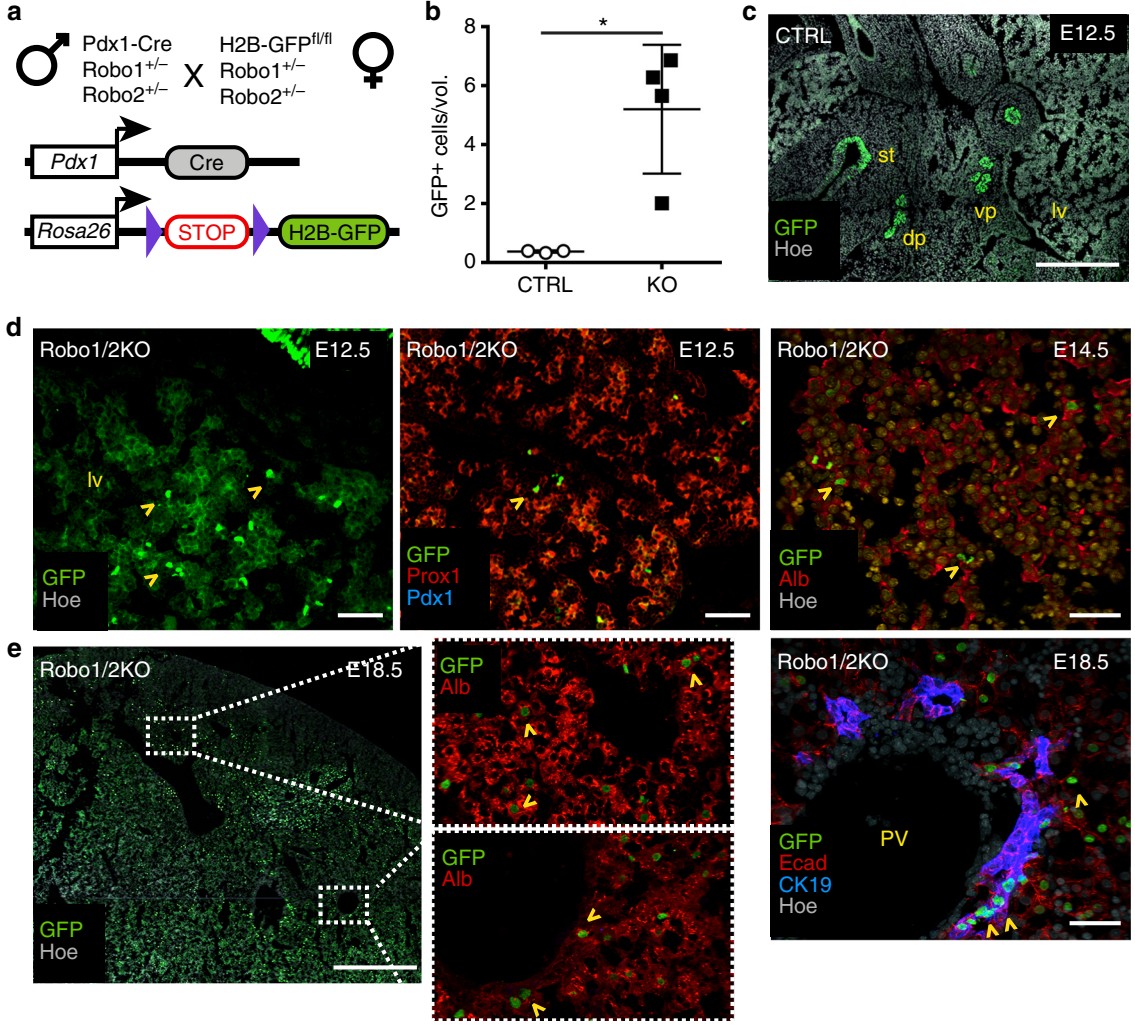

**Fig. 5** Lineage tracing of *Pdx1*-expressing progenitors in Robo1/2 KO embryos. **a** Schematic representation of the transgenes and breeding scheme used for lineage tracing experiments in Robo1/2 KO embryos. **b** Quantification of GFP+ cells in the E12.5 liver of CTRL ($n = 3$) and Robo1/2 KO ($n = 4$) embryos. Number of GFP+ cells was normalised to the liver bud volume. Error bars represent ± s.e.m. Two-tailed Student's *t*-test *$P < 0.05$. **c** Representative tiled IF image of E12.5 Pdx1-Cre;H2B-GFP;Robo1+/+/Robo2+/+ (CTRL) cryosections stained for GFP and Hoechst (Hoe) nuclear counterstain. As expected, cells in ventral pancreas (vp), dorsal pancreas (dp), stomach (st) and duodenum were labelled by Pdx1-Cre[24,29]. Scale bars, 200 μm. **d** Representative IF images of E12.5 and E14.5 Pdx1-Cre;H2B-GFP;Robo1/2 KO (Robo1/2 KO) cryosections stained with indicated antibodies and Hoechst (Hoe) as nuclear counterstain. Arrowheads indicate GFP+ cells in the liver (lv), which are also Prox1+ or Albumin+ (Alb). **e** Representative IF images of E18.5 Pdx1-Cre;H2B-GFP;Robo1/2 KO (Robo1/2 KO) cryosections stained with indicated antibodies and Hoechst (Hoe) as nuclear counterstain. Arrowheads indicate GFP/Albumin (Alb)-double positive cells in the hepatic parenchyma and GFP/CK19-double positive cells in the portal vein (PV) space. Scale bars, 50 μm

*Tead2*, were found in mouse early pancreatic progenitors[19]. To directly examine the consequences of *Robo1* and *Robo*2 deletion on Tead transcription factors in ventral pancreatic progenitors, we performed IF staining on E10.5 Robo1/2 KO embryos and measured the fluorescence intensity (Fig. 7b, c). Importantly, Tead levels were reduced specifically in Robo1/2 KO ventral pancreatic bud compared to controls, while the signal intensity was unchanged in dorsal pancreatic rudiments at this early stage (Fig. 7d, f).

Phosphorylation by the Hippo kinase cascade leads to cytoplasmic translocation and inactivation of YAP, while the un-phosphorylated YAP localises to the nucleus where it associates with Tead transcription factors[36,40,41]. Interestingly, using an antibody specific to the un-phosphorylated (active) form of YAP1, we found that E10.5 pancreatic progenitors displayed robust active-YAP nuclear localisation, whereas in the Robo1/2 KO pancreatic rudiment cells were mostly devoid of active-YAP

(Fig. 7e). Similarly, in Robo$^{PaΔ}$ conditional mutants as well as *in vivo* in mESC treated with Robo2-Fc, reduced Pdx1 levels correlated with diminished nuclear YAP localisation and Tead IF signal (Supplementary Fig. 6).

To further investigate such interplay between Robo and Tead factors, we assayed YAP/Tead transcriptional activity in response to the Slit/Robo pathway. To this aim, we used the well-established TEAD-binding element-driven luciferase reporter assay (8XGTIIC-luciferase) in HEK293T cells[43]. In line with the IF and RT-qPCR data, we found an induction of 8xGTIIC-Luc activity in cells transfected with *Robo2* construct; importantly, the induction was ligand dependent, being potentiated by the addition of Slit recombinant proteins to the medium (Fig. 7g). By contrast, overexpression of *Robo1* did not induce YAP/Tead-dependent transcriptional activity in this cellular context (Fig. 7g). Collectively, these results identified an interplay between Slit/Robo signalling and Tead transcription factors in pancreatic progenitors.

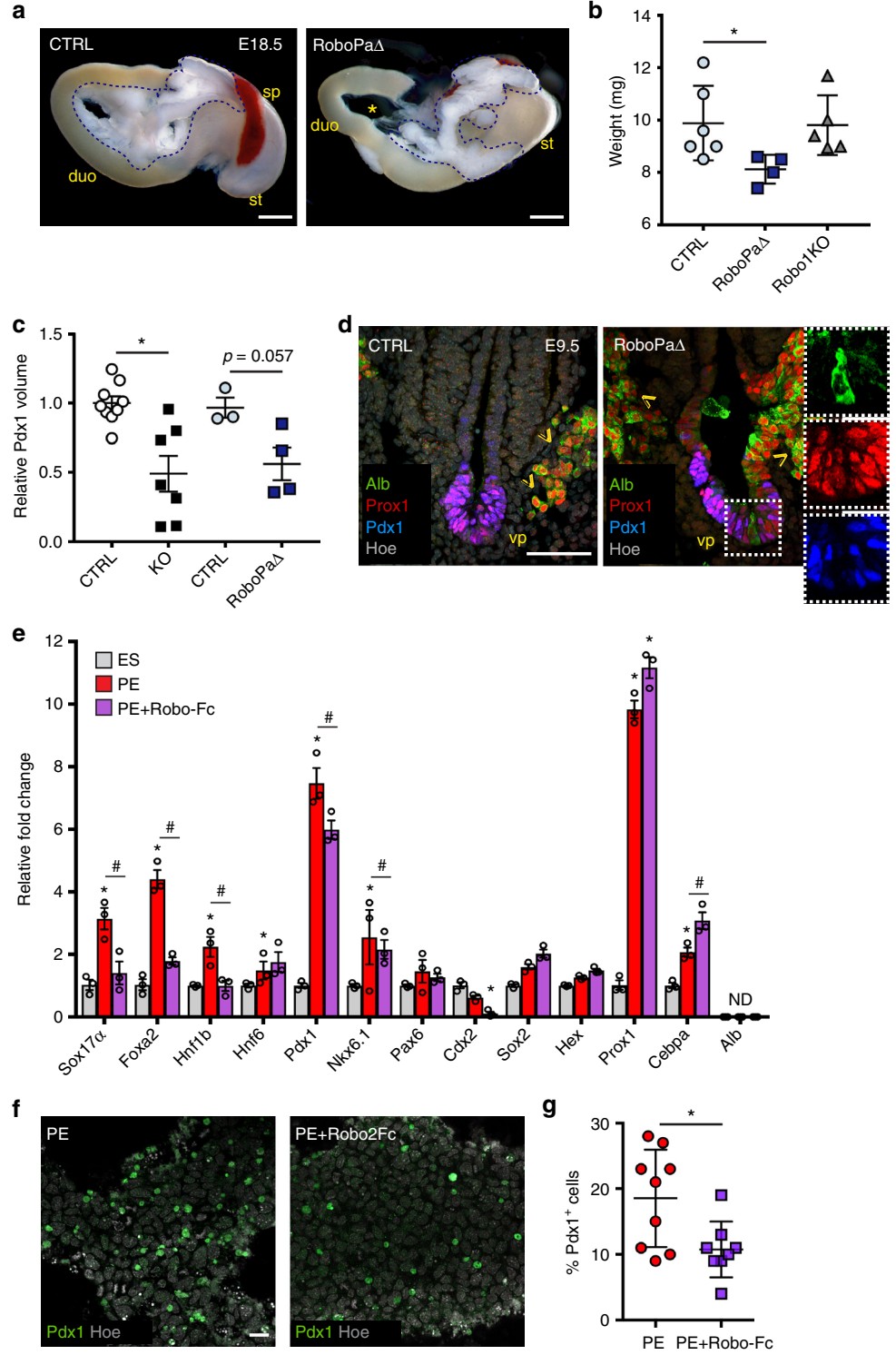

## Discussion

Altogether, our data provide evidence that Slit/Robo signalling regulates pancreas development both in vivo in the mouse and in ESCs. Specifically, we identified distinct activities of Robo receptors in pancreas progenitors: (i) maintenance of the newly specified pancreatic identity, (ii) regulation of cell death and (iii) of proliferation of pancreatic progenitors. Our analysis of the mutant phenotypes underscores temporally restricted activities of Robo within the pancreatic endoderm. In the absence of Robo receptors, soon after Pdx1-ventral pancreatic progenitors are specified at E9.5, we observed a destabilisation of pancreatic cell identity, which is concomitant with increased apoptosis. This suggests that a subset of cells, undergoing loss of identity, might be eliminated by cell death. By contrast, defects in cell proliferation become evident only later, starting from E10.5 onward in Robo mutant ventral pancreas, and subsequently affect both ventral and dorsal pancreas. Based on these findings we propose that these distinct embryonic activities together influence the final organ size, supporting the current view of pancreas organ size control[22]. Moreover, we identified here an interaction between

**Fig. 6** Robo signalling activity is restricted to pancreatic progenitors. **a** Pancreas gross morphology at E18.5. Robo$^{Pa\Delta}$ (Pdx1-Cre;Robo1$^{-/-}$;Robo2$^{fl/fl}$) shows pancreatic hypoplasia. Asterisk (*) indicates the head of the pancreas, which is mostly affected. Blue dotted line delineates the whole pancreas; duo duodenum, sp spleen, st stomach. Scale bars, 1 mm. **b** Quantification of pancreas weight in control (CTRL; $n = 6$), Robo$^{Pa\Delta}$ ($n = 4$) and single Robo1 KO ($n = 5$) E18.5 mouse embryos. Error bars represent ± s.e.m. *Two-tailed Student's $t$-test $P < 0.05$. **c** Conditional inactivation of *Robo* genes in pancreatic progenitors recapitulates the pancreatic size defect of Robo1/2 KO pancreas at E9.5. Measurement of Pdx1$^+$ ventral pancreatic bud volume ($\mu m^3$) in E9.5 CTRL, Robo1/2 KO and Robo$^{Pa\Delta}$ embryos. Pdx1$^+$ vp volume measurement was performed using the surface detection tool in Imaris. Error bars represent ± s.e.m. *Two-tailed Student's $t$-test $P < 0.05$. **d** Representative IF images of E9.5 CTRL and Robo$^{Pa\Delta}$ cryosections stained for Albumin (Alb), Pdx1 and Prox1. Hoechst (Hoe) was used as nuclear counterstain. Arrowheads indicate Prox1$^+$/Alb$^+$ liver progenitors in hepatic cords. In dashed box, Pdx1$^-$, Prox1$^+$ and Alb$^+$ cells within the ventral pancreatic epithelium (vp) of KO embryos. Insets show split channels of boxed area at higher magnification. Scale bars, 50 $\mu m$. **e** RT-qPCR analysis evaluating pancreatic gene expression in mESC differentiation into pancreatic endoderm (PE) in the absence or presence of Robo2-Fc. Data are represented as relative fold change. Values shown are mean ± s.e.m. ($n = 3$). *$P < 0.05$, two-tailed unpaired $t$-test for differentiated PE versus undifferentiated ESC; #$P < 0.05$ for differentiated PE versus PE+Robo2-Fc. ND not detected. **f** Representative IF images of PE and PE+Robo2-Fc stained with Pdx1 and Hoechst (Hoe) as nuclear counterstain. Scale bars, 20 $\mu m$. **g** Percentage of Pdx1$^+$ cells in PE and PE+Robo2-Fc cultures was calculated by counting Pdx1$^+$ cells relative to total number of Hoechst$^+$ nuclei per well. Error bars represent ±s.e.m. *Two-tailed Student's $t$-test $P < 0.05$

the Robo extrinsic signalling and the YAP/Tead transcriptional activity, which has not been reported in other cellular contexts. Our study suggests a model whereby Tead factors, downstream of Robo receptors, sustain the transcriptional programme in pancreatic progenitors and facilitate the expansion of pancreatic cells. A function for YAP and TEAD has been recently reported in human pancreatic differentiation[42], suggesting possible conservation also of the crosstalk with Slit/Robo extrinsic signalling.

Robo signalling has been classically linked to cytoskeleton organisation and morphogenesis[1,3,4,10,33]. Actually, a recent study reported that correct endocrine cell type sorting and proper islet architecture require the expression of *Robo* receptors in adult pancreatic β-cells[18]. It is therefore conceivable that Robo regulates morphogenesis, including cell adhesion and clustering, also in the pancreatic epithelium. Our initial analysis of the organisation of adherens junctions in Robo1/2 KO pancreata revealed reduced levels of E-cadherin in pancreatic progenitors, which are Prox1-high and Pdx1-negative. Eventually, Robo mutant cells form clusters that appear segregated and pushed out of the surrounding pancreatic epithelial sheet. This phenotype is reminiscent of the cellular processes used in *Drosophila* to eliminate misspecified cells from the developing imaginal disc epithelium (e.g., extrusion or cyst formation) or homoeostatic perturbations in metabolic activity (e.g., cell competition)[44–46]. Alternatively, it is conceivable that pancreatic cells, devoid of Robo receptors, acquire a migratory behaviour and actively leave the pancreatic epithelium. Indeed, our results show that Prox1-high and Pdx1-negative cells not only acquire hepatic gene expression, but also morphological features, which are reminiscent of embryonic liver cells[26,27,47]. Specifically, beside the reduced levels of E-cadherin, mutant Prox1-high cells display (i) changes in F-actin localisation, which accumulates at cell's leading edge instead of the apical localisation typical in pancreatic bud cells[48,49], (ii) laminin breakdown at the basement membrane and (iii) ectopic induction of *Hex* expression. During development, this set of events is required for the initiation of hepatoblast delamination and migration in the adjacent mesenchyme; previous work has shown that in the absence of *Prox1* or *Hex* in the mouse, hepatic endoderm fails to migrate and to preserve cell differentiation[26,27,47]. Hence, our findings suggest that Robo signalling in the ventral foregut is upstream of a transcriptional programme, which concomitantly favours pancreatic cell identity and prevents cell migration.

Robo functions in progenitor cells or stem cell niches are often mediated by transcriptional regulation[4,7,11,12]. For instance, in the *Drosophila* intestinal stem cell lineage, Robo2 has been identified as an upstream regulator of the transcription factor *Prospero*, forming a signalling loop that functions to maintain the enteroendocrine lineage[11]. On the other hand, in the mouse developing brain, Robo1 and Robo2 receptors have been reported to

cooperate with Notch signalling through transcriptional regulation of *Hes1*, favoring self-renewal and expansion of the neural progenitor pool[12]. Our study identifies a function for Slit/Robo pathway in preserving and expanding the pancreatic progenitor pool. In this cellular context, our results suggest that the Tead transcription factors is a downstream target of the Slit/Robo; the mechanisms through which this interaction occurs and whether it is a direct interaction remain to be elucidated. Recently, several studies reported mutual regulatory mechanisms between F-actin and Hippo/YAP, whereby actin cytoskeleton regulates the transcriptional activity of YAP by directing its subcellular localisation[40,41,43]. Thus, the F-actin cytoskeleton might also represent the link, mediating the crosstalk between the two pathways, in this context. Specifically, the changes in F-actin organisation, which occur in Robo mutant pancreatic cells, might represent the regulatory input for reducing active-YAP in the nuclei and, consequently, Tead transcriptional activity.

Beyond developmental implications, our findings will prove useful for comprehending the regulatory defects that drive pancreatic cancer. Since loss of cell identity is associated with PDAC[50–55], it is possible that cell-autonomous dysregulation of Robo contributes to cancer by recapitulating the embryonic phenotype that we reported here. Further investigation on Slit and its receptors Robo in pancreatogenesis will provide insights into the mutations identified in patients and their impact on the processes that lead to tumour formation as well as hold promise for therapeutic targets and early detection strategies.

## Methods

**Mouse strains.** Robo1$^{tm1Matl}$ (Robo1 KO), Robo2$^{tm1Mrt}$ (Robo2 KO) and Robo2$^{tm1Rilm}$ (Robo2-Flox) mouse strains have been previously described[3,7,8]. Robo1/2 KO were generated by intercrossing parents heterozygous for both mutant alleles. For conditional ablation in the pancreas, Robo2-Flox mice were intercrossed with Tg(Ipf1-cre)$^{1Tuv/Nci}$ (a.k.a. Pdx1-Cre) mice[29] and with Robo1$^{tm1Matl}$ (Robo1 KO)[3]. Briefly, mice with Robo2-Flox allele linked to Robo1$^-$ allele were crossed to mice carrying the same heterozygous allele and a Pdx1-Cre allele. For lineage tracing experiments, Robo1$^{+/-}$;Robo2$^{+/-}$ heterozygous mice were intercrossed with Pdx1-Cre and R26R-H2B-EGFP mice[28]. All animal experimentation was conducted in accordance with the local ethics committee (LaGeSO) for animal care.

**Cell culture.** Mouse ES cells (R1 line) were maintained on gelatin-coated plates with mouse embryonic fibroblasts (MEFs) in standard mESC medium: Dulbecco's modified Eagle's medium (Invitrogen), 2 mM glutamax (Invitrogen), 1 mM sodium pyruvate (Invitrogen), 0.1 mM nonessential amino acids (Invitrogen), 15 % foetal bovine serum (FBS) (PAN Biotech), 0.1 mM β-mercaptoethanol (Sigma) and 1000 U/mL leukaemia inhibitory factor (ESGRO). For differentiation assay, cultures were MEF depleted and seeded in mESC medium at high confluency on gelatin-coated dishes. For IF analysis, mESCs were plated on 8-well chamber slides (Ibidi) coated with gelatin. Monolayer differentiation was carried out using a well-established protocol[30,31]. Briefly, DE medium to day 2 consisted of RPMI medium (Invitrogen) and 0.2% FBS supplemented with 50 ng/mL Activin A and 25 ng/mL Wnt3a at day 1 and Activin A only at day 2. PE medium to day 8 consisted of

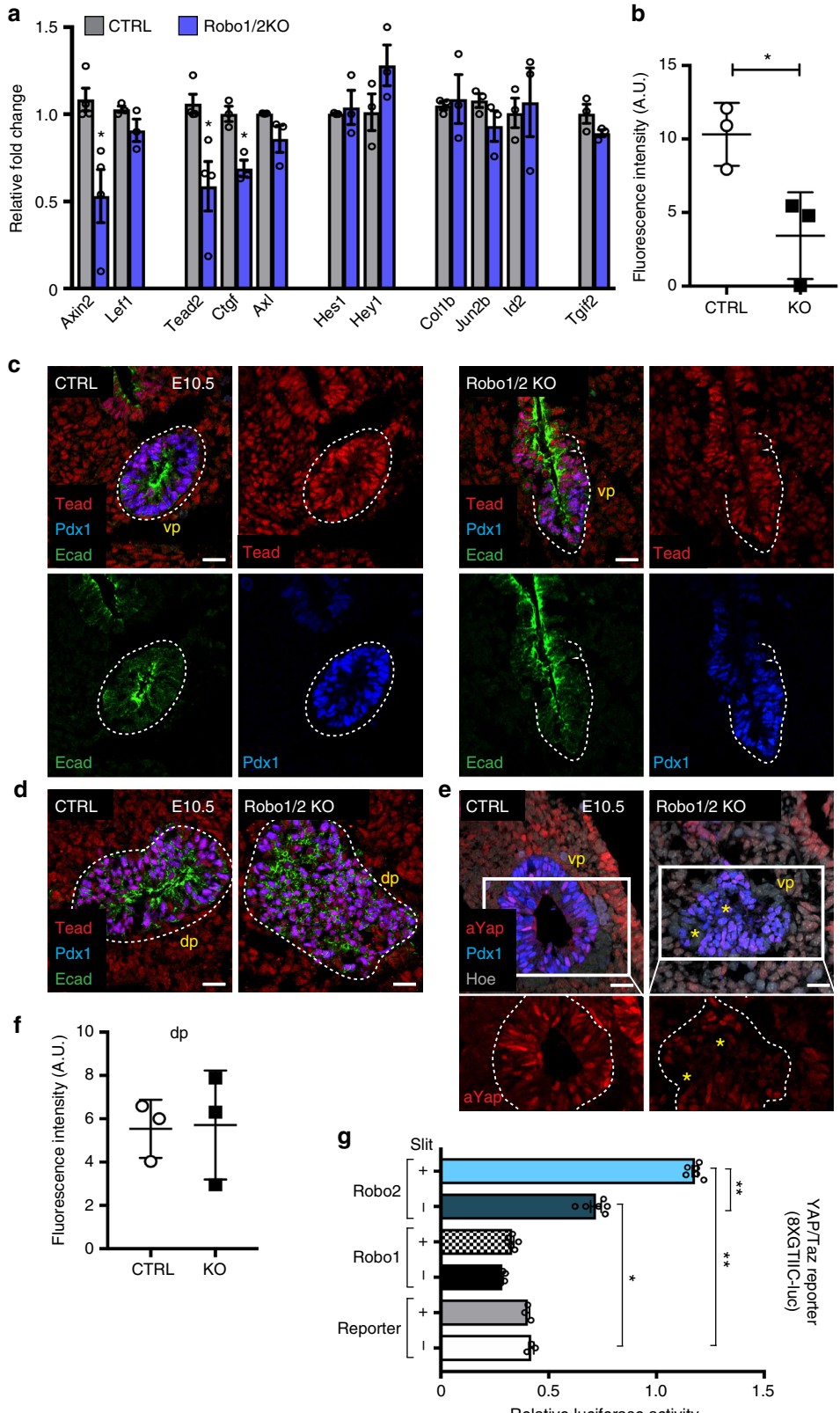

RPMI medium and 2% FBS supplemented with 3 ng/mL Wnt3a and 50 ng/mL FGF10. To block Robo signalling, recombinant mouse Robo2-Fc chimera protein (1 μg/mL) was added at DE stage. All recombinant proteins were purchased from R&D System unless otherwise stated.

**Immunohistochemistry and in situ hybridisation**. Mouse embryos and pancreata were fixed in 4% paraformaldehyde at 4 °C from 2 h to overnight. For whole-mount

immunostainings, fixed mouse embryos were incubated in freshly prepared PBSMT blocking solution (2% milk powder, 0.5% Triton X-100 in 1× phosphate-buffered saline (PBS)), and afterwards with primary antibodies at the appropriate dilution ON at 4 °C (see Supplementary Table 1). After extensive washes in fresh PBSMT solution at least 5–8 times, the embryos were incubated overnight with secondary antibodies at 4 °C in PBSMT solution[19]. Whole mouse embryos from E 9.5 onward were cleared in methyl salicylate for confocal microscope imaging. For cryosectioning, samples were equilibrated in 20% sucrose solution and embedded

**Fig. 7** Robo signalling influences YAP/TEAD transcription activity. **a** RT-qPCR analysis of indicated genes in E12.5 CTRL and Robo1/2 KO mouse pancreas. Data are represented as relative fold change. Values shown are mean ± s.e.m. ($n = 4$). *$P < 0.05$, two-tailed unpaired $t$-test. **b** Quantification of panTEAD fluorescence intensity in E10.5 CTRL and Robo1/2 KO ventral pancreas normalised to Hoechst intensity and area of epithelium. ($n = 3$). **$P < 0.01$, two-tailed unpaired $t$-test. **c** Representative IF images of E10.5 CTRL and Robo1/2 KO ventral pancreas (vp) cryosections stained for Pdx1, Tead and Ecad. Merge and single channels are shown. ($n = 3$). **d** Representative IF images of E10.5 CTRL and Robo1/2 KO dorsal pancreas (dp) cryosections stained for Pdx1, Tead and Ecad. Hoechst (Hoe) was used as nuclear counterstain. ($n = 3$). Dashed lines mark pancreatic epithelium (**c**, **d**). Scale bars, 20 μm. **e** Representative IF images of E10.5 CTRL and Robo1/2 KO ventral pancreas (vp) cryosections stained for active-Yap (a-Yap), Pdx1 and Hoechst (Hoe). Insets below show boxed area at higher magnification and a-Yap channel only. Dashed lines mark pancreatic epithelium. Asterisks indicate Pdx1-low cells that are devoid of a-Yap nuclear staining. ($n = 3$). Scale bars, 20 μm. **f** Quantification of panTEAD fluorescence intensity in E10.5 CTRL and Robo1/2 KO dorsal pancreas (dp) normalised to Hoechst intensity and area of epithelium. **g** Dual luciferase assay in HEK293 cells transfected with the 8xGTIIC-Lux reporter and *Renilla* luciferase reporter plasmids, in the presence (+) or absence (−) of hRobo1-EGFP, hRobo2-EGFP, untreated or exposed to Slit recombinant proteins, as indicated. Results are expressed as the ratio of Firefly to *Renilla* luciferase activity. Bars are mean + s.d. **$P < 0.01$, *$P < 0.05$, two-tailed unpaired $t$-test. AU arbitrary units

in OCT compound (Sakura). Cryosections (10 μm) were incubated with TSA (Perkin Elmer) blocking buffer for 1 h at room temperature and afterwards with primary antibodies at the appropriate dilution (see Supplementary Table 1). If necessary, antigen retrieval was performed by boiling the slides for 20 min in citrate buffer (Dako). Hoechst 33342 counterstaining was used at a concentration of 250 ng/mL. Whole-mount and cryosections (10 μm) in situ hybridisation were performed as previously described[19]. Antisense Pdx1[19], Robo2, Robo1, Slit1, Slit2, Slit3 (gift of F. Bareyre)[2] and Hex[27] in situ probes were used. Images were acquired on Zeiss AxioObserver, Discovery and Zeiss LSM 700 laser scanning microscope. Zen software was used to create maximum confocal $z$-projections and Huygens software (SVI) and Imaris software were used for 3D volume measurement analysis of confocal $z$-stacks. Quantification of the fluorescence intensity and area (E-cadherin$^+$ pancreatic epithelium) were measured using Fiji software on confocal images. For single nuclear fluorescence intensity quantification, Prox1 and Pdx1 intensity values of cells within each embryo were measured and subsequently corrected by linear normalisation within each embryo. This resulted in the same overall dynamic range of fluorescence intensities between embryos, allowing for improved comparability. Pdx1$^+$ vp volume measurement was performed using the surface detection tool in Imaris. Experiments were repeated three times; one representative field of view is represented for each staining.

**Luciferase assay**. Luciferase assays were performed in HEK293T cells with the YAP/TAZ-responsive reporter 8xGTIIC-luciferase (a gift of Stefano Piccolo; Addgene plasmid #34615)[43]. Cells were transfected with the 8xGTIIC-luciferase reporter plasmid (1 μg) together with pTk-*Renilla* for normalisation (25 ng) and with hROBO2-EGFP (1 μg) or hROBO1-EGFP (1 μg) (gift of O. Rocks), where indicated. Cells were stimulated with a mix of Slit ligands (hSlit1 (100 ng/mL; R&D 5199-SL-050), mSlit2 (100 ng/mL; R&D 5444-SL-050), hSlit3 (100 ng/mL; R&D 9067-SL-050)) 24 h post transfection followed by lysis in Promega cell culture lysis buffer. Luciferase and *Renilla* activity was quantified using the dual reporter assay kit (Promega) according to the manufacturer's instructions using a TECAN Infinite 200 Pro-luminometer. Luciferase assay experiments were repeated three times on independent samples. The results shown represent the mean and s.e.m. of triplicates of one representative experiment. Measures are normalised and represented as ratio Firefly/*Renilla*.

**Reverse transcription and quantitative PCR**. For RNA isolation, adult and embryonic tissues were dissected and snap-frozen on dry ice and RNA was extracted with Trizol (Invitrogen) according to the manufacturer's instructions. The High Pure RNA Isolation Kit (Roche) was used for RNA extraction from cultured cells. Total RNA was processed for reverse transcription (RT) using Transcriptor First Strand complementary DNA (cDNA) Synthesis Kit (Roche). A mix of anchored-Oligo(dT)18 and random hexamer primers was used to generate the cDNA. Real-time PCR reactions were carried out using FastStart Essential DNA Green Master Mix on LightCycler 96 system (Roche). Succinate dehydrogenase (SDHA) or 36B4 were used as reference genes. Primer sequences are provided in Supplementary Table 2. Gene expression levels were determined by the $2^{-\Delta\Delta CT}$ method following normalisation to reference genes. RT-qPCR experiments were repeated at least three times with independent biological samples; technical triplicates were run for all samples and no RT and no template controls were included in all experiments.

**Statistical tests**. All results are expressed as mean ± s.e.m. Sample sizes of at least $n = 3$ were used for statistical analyses except where indicated. All experiments were repeated at least three times. The significance of differences between groups was evaluated with Student's $t$-test. $P < 0.05$ was considered statistically significant.

**Reporting summary**. Further information on research design is available in the Nature Research Reporting Summary linked to this article.

## Data availability
All data supporting the findings of this study are available from the corresponding author on reasonable request. A source data underlying Figs. 1b, 6e, 7a and Supplementary Fig. 1c are provided as a Source Data file. A reporting summary for this Article is available as a Supplementary Information file.

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

## Acknowledgements

We thank all the members of the Spagnoli laboratory for their useful comments, suggestions and help. We thank W. Andrews (UCL, UK), W. Lu (Boston University, USA) and A. Chedotal (I. de la Vision, Paris, France) for sharing and shipping of the Robo2^tm1Rilm^ mouse strain. We thank I. Rooman (VUB, Belgium) for helpful discussions on the study. We thank L.G. Spagnoli and M. Mungo (Villa Pia and Histocytoservice, Rome, Italy) for human pancreatic tissues. We thank S. Piccolo (University of Padova, Italy) for providing us with the 8xGTIIC-luc, O. Rocks (MDC, Berlin, Germany) for hROBO2-EGFP, and hROBO1-EGFP and F. Bareyre (LMU, Munich, Germany) for Slit1, 2, 3 plasmids. This research was supported by funds from the Helmholtz Association and a grant from the GIF (I-1308-203), the German-Israeli Foundation for Scientific Research and Development. The F.M.S. lab. was supported by the ERC-POC grant (TheLiRep #641036), BIH (Tr. PhD grant) and EFSD/AZ grants.

## Author contributions

F.M.S. conceived the study, analysed data and wrote the manuscript. S.E. conceived the study, performed experiments, analysed data and wrote the manuscript. D.W. and H.N. performed and analysed experiments; S.D.F. performed lineage tracing experiments under the supervision of S.E. All authors approved the final version of the manuscript.
