## [Peer Review File · Nature Communications]

Reviewers' Comments:

Reviewer #1:

Remarks to the Author:

Review of Escot et al.

Robo signaling pathway functions as a gatekeeper of pancreatic identity. (*italics added*)

Summary of findings as presented:

1. Robo1, Robo2 and Slit3 have organ-specific patterning.
2. Loss of both Robo1/2 leads to reduction in pancreas size (particularly head of the pancreas, which comes from ventral pancreas).
3. Robo1/2 KO leads to less stable identity between pancreas and liver.
 - a. Ventral pancreas has liver-like cells (Prox1+, Pdx-, Alb+, Afp+)
 - b. They see increased GFP+ cells in the liver (with PDX-Cre driver).

Their explanation for this last point:

"Taken together, these results suggested that in the absence of Robo1 and Robo2 a sub-set of pancreatic progenitors lose their original identity and switch to a liver cell fate, being eventually misallocated to the liver bud." <- Did the cells migrate there? This seems like it could be an over-reach given the presented data.

4. Changes/impairment in pancreatic gene expression when mESC differentiations are treated with Robo-Fc (Robo2-Fc chimera that should block Robo/Slit signaling).

These changes are unconvincing frankly... would prefer seeing staining/flow cytometry.

5. Reduced TEAD activity in PDX1+ cells in Robo1/2 KO ventral pancreas, but not in dorsal pancreas.

This is pretty 'weak' given the images shown in Fig 6c). Panel 6b should display Dorsal pancreas quantification.

6. They show YAP/Taz reporter can be activated by combined over-expression of Slit3/Robo2 (Fig 6e).

Overall a good 'on-target' paper. The connection to PDAC (in the intro/discussion) is a bit tenuous—could it be expanded or removed? The developmental results are quite convincing.

Reviewer #2:

Remarks to the Author:

The manuscript by Escot and Spagnoli et al. addressed a role of the Slit/Robo signaling pathway in pancreatic organogenesis using the mouse system. In the course of the mouse embryonic development, the Robo1 and Robo2 genes as well as the Slit3 ligand gene was expressed in a pattern reminiscent of their possible involvement in formation of pancreatic buds, in particular the ventral one. Analyses of Robo1 and Robo2 double knockout (Robo1/2 KO) mice revealed that these receptors were important for preserving pancreatic progenitor identity. A lineage tracing approach confirmed that those pancreatic progenitors that lost the Robo1/2 expression underwent an aberrant lineage conversion and gained a hepatoblast identity. Finally, as a potential downstream mechanism, the authors proposed a role of the YAP/Tead transcriptional activity. Overall, this is a very interesting study reporting a novel signaling pathway critically involved in pancreatic development. The phenotype of the Robo1/2 KO mice is remarkable, and the issue on the role of the Robo signaling pathway in the maintenance of the pancreatic lineage identity was well supported by the genetic fate tracing experiment. In contrast, however, the underlying mechanisms whereby the Slit/Robo signaling regulates the lineage identity seems to have been less sufficiently characterized. The link between the Slit/Robo signaling and the YAP/Tead transcriptional activity remains ambiguous and should be more strengthened with additional evidence. It may be plausible to assume a possibility that actin cytoskeleton reorganization is

involved in connecting these signaling units and should be examined.

The same authors' group has previously published a very elegant study regarding the role of the transcriptional regulator TGIF2 in hepatic–pancreatic lineage determination and conversion (Cerdeira-Esteban, Nat Commun 2017). The relationship between this molecule and the Slit/Robo–YAP/Tead axis identified herein should also be addressed and presented.

Specific comments:

- Page 4, line 19. The authors state that "Robo1 expression was detected in both dorsal and ventral pancreatic buds as well as liver bud". This appears contradictory to the data shown in Fig. 1b, which indicates that the expression pattern of Robo1 is essentially the same as that of Robo2 at E10.5. In Fig. 1b, the data should be presented so that the gene expression levels at E10.5 and E14.5 can be compared.

- In the Robo1/2 KO embryo, the ectopic Pdx1-low/negative Prox1-high pancreatic cells appeared to delaminate from the pancreatic epithelium and migrate outward, reminiscent of the developmental process of liver bud formation. As Hhex is known to be essential for the delamination and migration of early hepatoblasts from the hepatic diverticulum in formation of the liver bud, expression of this transcription factor should also be examined. Can the authors provide any data as to whether those ectopic cells actively delaminated and migrated, or rather passively eliminated, from the epithelium?

- While the phenotype of the Robo1/2 KO embryos suggest that the Slit/Robo signaling is involved in (i) maintenance of cell identity, (ii) suppression of cell death, and (iii) promotion of cell proliferation, in pancreatic progenitor cells, it remain unclear whether and how these three cellular processes are related with each other. The authors should address this issue. For instance, what happens if induction of apoptosis is blocked in the Robo1/2 KO embryos? Does this affect maintenance of cell identity and/or cell proliferation, and further lead to a partial rescue of the phenotype of pancreatic organ size reduction? The mouse ESC ex vivo system (Fig. 5d) should be employed to examine the effect of the blockade of the Slit/Robo signaling on apoptosis and proliferation of pancreatic progenitor cells.

- Fig. 3, f and g. Phospho-histone H3 is assumed to be a marker for cells undergoing mitosis (i.e., the G2-to-M phase transition). The authors should also examine other markers for cell proliferation and/or cell cycle progression, such as Ki67 and BrdU (or EdU) incorporation, to substantiate the notion that the loss of Robo receptors leads to reduced proliferation.

- Fig. 4. The authors showed the presence of ectopic GFP-positive cells in the livers of Robo1/2 KO embryos, and also of the control embryos albeit to a lesser extent. Are these Pdx1-Cre lineage labeled cells persistent and present until at the adult stage in the liver (in the Robo1/2 KO and the control mice)?

- Fig. 4a. The corresponding data for alpha-fetoprotein should also be included (perhaps as a supplementary material).

- Fig. 5d. Expression of the hepatoblast markers albumin, alpha-fetoprotein, and Prox1 should also be examined.

- With regard to the suggested role of the YAP/Tead pathway functioning downstream of the Slit/Robo signaling (Fig. 6), it should be determined whether YAP activation is indeed under the control of the Slit/Robo signaling in vivo (such as by immunostaining for YAP localization) and in vitro (Fig. 6e). In the mouse ESC ex vivo system experiment (Fig. 5d), can forced activation of Tead transcriptional activity, or of the upstream YAP signaling, rescue the effect of the blockade of the Slit/Robo signaling?

Reviewer #3:

Remarks to the Author:

Escot and collaborators have studied the development of the pancreas in Robo1/Robo2 knockout mice. Their data suggest that Robo receptors control the proliferation, differentiation and survival of pancreatic cells. The work is potentially interesting but also looks quite immature with many experiments lacking. At the end, one does not know what is their working model.

The involvement of Slit ligands is completely neglected. Why did the authors only studied Slit3 expression without pushing this farther? They should also show Slit1 and Slit2 as they all bind to Robo1/2 receptors. Slit knockouts are also available and if the authors favor Slit3 they should be able to study pancreas in Slit3^{-/-} mice.

The authors only used Robo1 and Robo2 antibodies to study PDAC (Fig S1) . Could they use the antibodies to confirm the mRNA data ? Why do Robo1 and Robo2 have redundant function if they are expressed in non overlapping territories ? is Robo1 upregulated in Robo2 KO and vice versa ? How does the pancreas of Pdx1-Cre ;Robo2lox or Robo1 single KO looks like ?

Obviously, one needs to know whether Pdx1-CreRobo1/Robo2lox/lox (Robo1/2Pdelta) mice are viable and develop PDAC or other pancreatic diseases.

Why did the authors show Robo1/Robo2 expression in PDAC ? As is, this has nothing to do with the paper unless the authors show us pancreas structure in adult Robo1/2Pdelta mice.

The analysis of the Robo1/2Pdelta is essential to confirm that the defects observed in Robo1/2 null are due to Robo function in the pancreas. However, the authors do not compare the phenotype of both lines just one picture and the weight) and one cannot tell from the data if they fully phenocopy each other.

They also don't show if Robo2 is eliminated from all pancreatic cells in Robo1/2Pdelta mutants. Robo2 in situ or immuno should be performed to assess this.

Fig1c among others : the reduction of the pancreas is not obvious at all for non specialists. Could they perform whole-mount staining of the samples with a pancreatic marker such as Pdx1?

Likewise, the pancreas contours should be delineated on Fig S2.

Where is sox17 normally expressed compared to Pdx1 ? again they should not assume that all readers know pancreatic development.

Figure 2a : the authors only show sections but then talk about pancreas volume. Did they image and quantified all sections ? Here again, whole-mount staining would be a real plus.

The caspase data (Fig3A) are not very convincing and they should show Caspase3 staining alone (low mag) in addition to the merge. The quantification should also be by square micrometers (or cubic micrometers) and not "by area" (what is the size of the area ? the legend does not help).

The mouse ESC data have no interest (they could for instance have silenced Robo expression or added Slit molecules) and should be either better used (more experiments) or deleted. Adding Robo-Fc does not tell much of Slit/Robo signaling unless they show that some Slits are present in the supernatant.

How could Robo and YAP/TEAD be linked molecularly to Robo? No mechanism is proposed or tested. Again the data were just obtained using Robo1/2 null embryos but not with the Robo1/2Pdelta embryos.

Other points

"loxP-flanked Robo2 allele and the Pdx1-Cre"transgenic strain in a Robo1-deficient background^{8, 26,} I did not find the description of this double mutant line and its validation in the two references (just on the single KO). How was it generated and what was the source of the Robo1 mutant?

Is cre only expressed in the pancreas in Pdx1-Cre mice? is GFP detected outside the pancreas in Pdx1-Cre;R26RH2B-GFP embryos (this could also be genetic background dependent). Only E12.5 is shown but later stages should also be studied to confirm the specificity of Cre expression.

Supplementary Figure 4b, c: the data are not quantified and from the pictures, the number of axons seems much higher in the mutant, and it is impossible to tell anything from the PECAM staining.

"Tead levels were reduced specifically in Robo1/2 KO ventral pancreatic bud compared to controls, while the signal intensity was unchanged in dorsal pancreatic rudiments at this early stage (Fig. 6b-d)." This is basically impossible to tell from the data. Where are dorsal and ventral? Moreover, Tead expression seems to be decrease everywhere on Fig 6c.

Why a mix of Slit1, Slit2 and Slit3 was added to the 203 cells for the luciferase assay? How was the optimal concentration determined (why 300 ng/ml? were lower doses tested?). Are the same results obtained with Robo1?

Ref 47 is incorrect : 47. Prévot Pea.???

Response to reviewer comments

We thank the reviewers for their insightful and helpful comments, as we feel addressing them has improved our manuscript significantly. We have carried out new experiments (which have added five sets of panels to our existing figures, one new figure and two new Supplementary figures) and we have revised the text to address the concerns. Below is a point-by-point response to reviewers' concerns with our responses shown in blue. We hope the reviewers and editor will find the revised manuscript now suitable for publication at *Nature Communications*.

Reviewer #1

Summary of findings as presented:

1. Robo1, Robo2 and Slit3 have organ-specific patterning.
2. Loss of both Robo1/2 leads to reduction in pancreas size (particularly head of the pancreas, which comes from ventral pancreas).
3. Robo1/2 KO leads to less stable identity between pancreas and liver.
 - a. Ventral pancreas has liver-like cells (Prox1+, Pdx-, Alb+, Afp+)
 - b. They see increased GFP+ cells in the liver (with PDX-Cre driver).

Their explanation for this last point:

"Taken together, these results suggested that in the absence of Robo1 and Robo2 a sub-set of pancreatic progenitors lose their original identity and switch to a liver cell fate, being eventually misallocated to the liver bud." <- Did the cells migrate there? This seems like it could be an over-reach given the presented data.

We agree and thank the reviewer for pointing this out. We now provide images of E9.5 CTRL and Robo1/2 KO (see new Fig 4) showing morphological features of mutant cells, including reduced E-cadherin, F-actin reorganization and laminin breakdown, which are reminiscent of migrating hepatoblasts. We also refer to these data on page 8 of the Results Section and in the new Discussion section. Our new findings suggest that Robo signaling is upstream of a transcriptional network in the ventral foregut to preserve a transcriptional program, which concomitantly favors pancreatic cell identity and prevents cell migration.

Nevertheless, to avoid any overstatement, we have removed the sentence "being eventually misallocated to the liver bud."

4. Changes/impairment in pancreatic gene expression when mESC differentiations are treated with Robo-Fc (Robo2-Fc chimera that should block Robo/Slit signaling). These changes are unconvincing frankly... would prefer seeing staining/flow cytometry.

We thank the reviewer for this suggestion. We now included immunofluorescence images for Pdx1 and Nkx6.1 in mES cultures differentiated into pancreatic endoderm. In line with the RT-qPCR results, we found significant reduction of Pdx1-positive cells upon treatment of the differentiating cells with recombinant Robo2-Fc chimera for blocking Robo signaling. We also further explained the biological activity of Robo2-Fc

chimera in other cellular contexts, included additional citations and the rationale for using in mESC undergoing endoderm differentiation (see page 9 of Results Section, new Figure 6 and Supplementary Fig. 6).

5. Reduced TEAD activity in PDX1+ cells in Robo1/2 KO ventral pancreas, but not in dorsal pancreas.

This is pretty 'weak' given the images shown in Fig 6c). Panel 6b should display Dorsal pancreas quantification.

We have included the quantification of the Fluorescence intensity levels in dorsal pancreas (see new Figure 7d,f) and have replaced the Tead immunofluorescence images and included E-cadherin as "internal control" for the quality of the staining in the epithelium. This is now shown in new Figure 7 and Supplementary Fig. 6.

6. They show YAP/Taz reporter can be activated by combined over-expression of Slit3/Robo2 (Fig 6e).

Overall a good 'on-target' paper. The connection to PDAC (in the intro/discussion) is a bit tenuous— could it be expanded or removed? The developmental results are quite convincing.

We thank the reviewer for the support. We have now reduced the section about PDAC in the Introduction as well as Discussion and removed the IHC of PDAC tissue.

Reviewer #2

The manuscript by Escot and Spagnoli et al. addressed a role of the Slit/Robo signaling pathway in pancreatic organogenesis using the mouse system. In the course of the mouse embryonic development, the Robo1 and Robo2 genes as well as the Slit3 ligand gene was expressed in a pattern reminiscent of their possible involvement in formation of pancreatic buds, in particular the ventral one. Analyses of Robo1 and Robo2 double knockout (Robo1/2 KO) mice revealed that these receptors were important for preserving pancreatic progenitor identity. A lineage tracing approach confirmed that those pancreatic progenitors that lost the Robo1/2 expression underwent an aberrant lineage conversion and gained a hepatoblast identity. Finally, as a potential downstream mechanism, the authors proposed a role of the YAP/Tead transcriptional activity.

Overall, this is a very interesting study reporting a novel signaling pathway critically involved in pancreatic development. The phenotype of the Robo1/2 KO mice is remarkable, and the issue on the role of the Robo signaling pathway in the maintenance of the pancreatic lineage identity was well supported by the genetic fate tracing experiment. In contrast, however, the underlying mechanisms whereby the Slit/Robo signaling regulates the lineage identity seems to have been less sufficiently characterized. The link between the Slit/Robo signaling and the YAP/Tead transcriptional activity remains ambiguous and should be more strengthened with additional

evidence. It may be plausible to assume a possibility that **actin cytoskeleton reorganization is involved in connecting these signaling units and should be examined.**

We thank the reviewer for this very important suggestion. In compliance with the reviewer request, we have examined F-actin cytoskeleton using actin fluorescent probes (488-phalloidin) and confocal imaging analysis in E9.5 CTRL and Robo1/2 KO. We now provide these images in new Fig. 4 showing that Pdx1-low / Prox1-high mutant pancreatic cells display changes in F-actin distribution, being not anymore confined to the apical domain but visible on the lateral and basal surfaces. Overall, the morphological features displayed by the mutant cells, including reduced E-cadherin, F-actin reorganization and laminin breakdown, are reminiscent of migrating hepatoblasts (see also below our answer to Point 2 of the same reviewer). We refer to these new data in page 8 of the Results section and in the new Discussion section.

- The same authors' group has previously published a very elegant study regarding the role of the transcriptional regulator TGIF2 in hepatic-pancreatic lineage determination and conversion (Cerdeira-Esteban, Nat Commun 2017). The relationship between this molecule and the Slit/Robo-YAP/Tead axis identified herein should also be addressed and presented.

In compliance with the reviewer, we have checked *Tgif2* expression in Robo1/2 mutant pancreas and found no modulation (see new Figure 7a), suggesting that Robo signaling is not upstream of *Tgif2* induction.

Specific comments:

1) Page 4, line 19. The authors state that "Robo1 expression was detected in both dorsal and ventral pancreatic buds as well as liver bud". This appears contradictory to the data shown in Fig. 1b, which indicates that the expression pattern of Robo1 is essentially the same as that of Robo2 at E10.5. In Fig. 1b, the data should be presented so that the gene expression levels at E10.5 and E14.5 can be compared.

We apologize with the reviewer if this sentence was misleading. This has been now edited and the same scale is used to compare gene expression levels at E10.5 and E14.5 (see new Figure 1 b).

2) In the Robo1/2 KO embryo, the ectopic Pdx1-low/negative Prox1-high pancreatic cells appeared to delaminate from the pancreatic epithelium and migrate outward, reminiscent of the developmental process of liver bud formation. As *Hhex* is known to be essential for the delamination and migration of early hepatoblasts from the hepatic diverticulum in formation of the liver bud, expression of this transcription factor should also be examined. Can the authors provide any data as to whether those ectopic cells actively delaminated and migrated, or rather passively eliminated, from the epithelium?

We thank the reviewer for pointing this out. In compliance with the reviewer request, we examined *Hhex* expression in the ventral endoderm in the

absence of Robo receptors and included here a thorough characterization of the morphological changes occurred in Robo mutant cells. The new data are now included in new Figure 4. Interestingly, in the absence of Robo receptors ventral pancreatic progenitors not only acquired expression of liver genes (high Prox1, Albumin, AFP) but also display morphological features typical of hepatoblasts undergoing delamination and migration (see also above). Based on these findings, we propose that Robo acts in the ventral foregut to preserve a transcriptional program, which concomitantly favors pancreatic cell identity and prevent cell migration. We discuss this on page 14 of the revised manuscript.

3) While the phenotype of the Robo1/2 KO embryos suggest that the Slit/Robo signaling is involved in (i) maintenance of cell identity, (ii) suppression of cell death, and (iii) promotion of cell proliferation, in pancreatic progenitor cells, it remain unclear whether and how these three cellular processes are related with each other. The authors should address this issue. For instance, what happens if induction of apoptosis is blocked in the Robo1/2 KO embryos? Does this affect maintenance of cell identity and/or cell proliferation, and further lead to a partial rescue of the phenotype of pancreatic organ size reduction?

The mouse ESC *ex vivo* system (Fig. 5d) should be employed to examine the effect of the blockade of the Slit/Robo signaling on apoptosis and proliferation of pancreatic progenitor cells.

We thank the reviewer for bringing up this important discussion point. In fact, various processes are affected by Robo signaling in pancreatic cells, but these activities are temporally-restricted. First, soon after Pdx1-ventral pancreatic progenitors are specified, we observed occurrence of “destabilization of pancreatic cell identity”, which is concomitant with increased apoptosis. This suggests that a subset of cells, undergoing loss of identity, might be eliminated by cell death. The increase in cell death is measured only at E9.5 in the ventral pancreatic territory. By contrast, the defects in cell proliferation become evident only later, starting from E10.5 onward, and subsequently affect both ventral and dorsal pancreas. Based on these findings, we propose that these distinct embryonic activities influence together the final organ size, supporting the current view of pancreas organ size. In the revised manuscript, we have included further examination of proliferation at additional time-points (E9.5, E10.5 and E12.5) and in different organ rudiments (vp, lv, dp) (see new Fig. 3 and new Supplementary Fig. 3). Moreover, this important aspect is now discussed in the new Discussion section.

Unfortunately, we have not found a way to experimentally block apoptosis that works *in vivo* in our mouse models. However, as suggested by the reviewer, we could confirm the effect on proliferation *ex vivo* in mESC cultures undergoing differentiation in the presence of Robo-Fc (see new Supplementary Fig. 6).

4) Fig. 3, f and g. Phospho-histone H3 is assumed to be a marker for cells undergoing mitosis (i.e., the G2-to-M phase transition). The authors should also examine other markers for cell proliferation and/or cell cycle progression,

such as Ki67 and BrdU (or EdU) incorporation, to substantiate the notion that the loss of Robo receptors leads to reduced proliferation.

It is correct that Ki67 is a good marker for all stages in the cell cycle, while pHH3 is more specific staining only cells in late G2 and mitosis. For this reason, it is logical to obtain more Ki67 stained cells than pHH3 stained cells from the same tissue sample. However, in our lab. IF staining with antibodies against Ki67 did not work in early stage mouse embryonic tissues (E8.5 – E12.5). We tried three different antibodies and various antigen retrieval procedures without any success, the same antibodies and conditions work for us in adult and later stages (see Cerda- Esteban et al. Nat Comms 2017). This is actually in line with old literature saying that Ki67 expression starts after E10.5 in the mouse (Mitsuyoshi et al. BBRC 1997 235, 191-196). Nevertheless, to reinforce our observation on cell proliferation, we characterized pHH3-positive cells at additional embryonic stages and in different progenitor compartments (see also answer to Point 3).

5) Fig. 4. The authors showed the presence of ectopic GFP-positive cells in the livers of Robo1/2 KO embryos, and also of the control embryos albeit to a lesser extent. Are these Pdx1-Cre lineage labeled cells persistent and present until at the adult stage in the liver (in the Robo1/2 KO and the control mice)?

We thank the reviewer for asking this important question. Pdx1-descendant cells are indeed persistent in the liver of the Robo1/2 KO (as well as of controls) newborn. We found GFP-positive cells that are positive for hepatocyte (Albumin) and biliary epithelial cell markers (CK19, Sox9). The latest time point we could examine is E18.5, because the Robo1/2 KO mutation is lethal right after birth (Long et al. 2004, Grieshammer et al. 2004, Domyan et al 2013). The lineage tracing results at additional time points (E14.5 and E18.5) are in new Figure 5.

6) Fig. 4a. The corresponding data for alpha-fetoprotein should also be included (perhaps as a supplementary material).

The immunofluorescence images showing alpha-fetoprotein within the ventral pancreatic epithelium has been added to the new Supplementary Figure 3.

7) Fig. 5d. Expression of the hepatoblast markers albumin, alpha-fetoprotein, and Prox1 should also be examined.

In compliance with the reviewer request, we have expanded the qPCR analysis of mESC experiment to include *Prox1* and additional hepatic markers. The differentiation protocol that we employ is directed to pancreatic lineage, therefore we normally do not observe induction of liver differentiation markers (D'Amour et al., Nostro et al. 2011, Rodriguez-Seguel et al. 2013). Consistently, we did not detect any induction of Albumin transcript. However, we found that transcription factors, such *Hex*, *Prox1*, *C/ebpa*, are all strongly induced upon inhibition of Robo signaling, which is in line with the *in vivo* observations.

8) With regard to the suggested role of the YAP/Tead pathway functioning downstream of the Slit/Robo signaling (Fig. 6), it should be determined whether YAP activation is indeed under the control of the Slit/Robo signaling *in vivo* (such as by immunostaining for YAP localization) and *in vitro* (Fig. 6e). In the mouse ESC *ex vivo* system experiment (Fig. 5d), can forced activation of Tead transcriptional activity, or of the upstream YAP signaling, rescue the effect of the blockade of the Slit/Robo signaling?

We thank the reviewer for this important suggestion. We have included in the revised manuscript IF analyses of the cellular localization of active Yap *in vivo* in the mouse ventral pancreas of Robo1/2 KO and Robo^{PaΔ} as well as in mESC exposed to Robo2-Fc. The new analysis shows reduced nuclear Yap in the absence of Robo signalling, which supports Tead IF and luciferase reporter results. The new data has been added in new Figure 6 and new Supplementary Figure 6.

Future studies with genetic tools are required to dissect the epistatic relationship between the two pathways. To date, we could not answer the question using transient methods or chemical compounds in ES cells undergoing differentiation.

Reviewer #3 (Remarks to the Author):

Escot and collaborators have studied the development of the pancreas in Robo1/Robo2 knockout mice. Their data suggest that Robo receptors control the proliferation, differentiation and survival of pancreatic cells. The work is potentially interesting but also looks quite immature with many experiments lacking. At the end, one does not know what is their working model.

The involvement of Slit ligands is completely neglected. Why did the authors only studied Slit3 expression without pushing this farther? They should also show Slit1 and Slit2 as they all bind to Robo1/2 receptors. Slit knockouts are also available and if the authors favor Slit3 they should be able to study pancreas in Slit3^{-/-} mice.

We apologize with the reviewer, if our text was misleading, we had actually examined all three Slit ligands in the mouse embryo at the relevant embryonic stages and found only Slit3 to be expressed in the mesenchyme surrounding the ventral pancreas. To avoid any confusion, we have now included the ISH images for Slit1 and Slit2 in the mouse embryo and qPCR in differentiating mESC in Supplementary Figure 1.

It is true that Slit3 mutant animals are available (Liu et al., 2003; Yuan et al., 2003) and they show absence of most of the mesenchyme surrounding the respiratory and digestive system. Thus, they deserve future investigation with regard to a pancreatic phenotype.

The authors only used Robo1 and Robo2 antibodies to study PDAC (Fig S1). Could they use the antibodies to confirm the mRNA data? Why do Robo1 and Robo2 have redundant function if they are expressed in non overlapping

territories ? is Robo1 upregulated in Robo2 KO and vice versa ?

We agree with the reviewer, it would have been much easier to use antibodies and perform IF staining instead of ISH. Unfortunately, all the antibodies that we have tested against the Robo receptors work in adult mouse tissues and human tissues (e.g. adult pancreatic islets), but do not in mouse embryonic tissues. Therefore, we used *in situ* hybridization, RT-qPCR and X-gal staining to study *Robo* genes in the mouse embryo. The following antibodies were tested: rabbit anti-ROBO2 (ab64158, Abcam); rabbit anti-ROBO2 (ab75014, Abcam); goat anti-ROBO2 (AF3147, R&D); rabbit anti-ROBO1 (ab7279, Abcam); goat anti-ROBO1 (AF1749, R&D).

Concerning the expression pattern, we now added a sentence to clarify this in the Result section. Briefly, both Robo1 and Robo2 receptors are expressed in ventral pancreas, where they could have redundant function, and both are expressed at very low level in dorsal pancreas and liver. In addition, Robo2 is specifically expressed in the foregut, already at E8.5.

Moreover, in compliance with the reviewer's request, we checked if *Robo1* is upregulated upon deletion of Robo2 by ISH. The ISH images are included in the new Supplementary Fig. 2, no induction was detected.

How does the pancreas of Pdx1-Cre ;Robo2lox or Robo1 single KO looks like ? Obviously, one needs to know whether Pdx1-CreRobo1/Robo2lox/lox (Robo1/2PAdelta) mice are viable and develop PDAC or other pancreatic diseases.

The single Robo1 KO and Robo2 KO embryos do not display pancreatic phenotype and are viable. These data are shown in Fig S2 and Fig. 6. Therefore, we conditionally inactivated Robo2 using Pdx1-Cre in a Robo1 mutant background.

Concerning the adult phenotype, our study focused on the embryonic function(s) of Robo genes in pancreatic progenitors and we did not analyze here the role of the pathways in the adult or in pancreatic diseases. Actually, a manuscript by Ilse Rooman and colleagues, which is submitted back-to-back with ours, addresses the biology of the pathway in pancreatic diseases.

Why did the authors show Robo1/Robo2 expression in PDAC ? As is, this has nothing to do with the paper unless the authors show us pancreas structure in adult Robo1/2PAdelta mice.

We agree with the reviewer and removed the IHC of Robo1 and Robo2 in human PDAC tissues, since our study focuses on the early developmental function of the Robo signaling in pancreatic progenitors.

The analysis of the Robo1/2PAdelta is essential to confirm that the defects observed in Robo1/2 null are due to Robo function in the pancreas. However, the author do not compare the phenotype of both lines just one picture and the weight) and one cannot tell from the data if they fully phenocopy each other. They also don't show if Robo2 is eliminated from all pancreatic cells in

Robo1/2PAdelta mutants. Robo2 in situ or immuno should be performed to assess this.

We thank the reviewer for this suggestion. We have now expanded the characterization of the Robo1/2PAdelta embryonic phenotypes, including maintenance of pancreatic identity and influence on Tead/Yap signaling at E9.5. The new data are included in new Figure 6 and new Supplementary Figure 6 and are consistent with the phenotype of Robo1/2 KO. In addition, we showed by ISH analysis that *Robo2* gene expression is ablated in Robo1/2PAdelta embryos (see new Supplementary Fig. 2). We apologize for not clarifying this in our previous submission, but we assumed that the Pdx1-Cre mouse strain used here is well characterized and commonly used in the field. The Pdx1-Cre activity and labelled cells are shown in Fig. 5c (see also below).

Fig1c among others : the reduction of the pancreas is not obvious at all for non specialists. Could they perform whole-mount staining of the samples with a pancreatic marker such as Pdx1? Likewise, the pancreas contours should be delineated on Fig S2.

Fig. 1c shows the gross morphology of the entire adult pancreas at birth. To help the reader, we now included in Supplementary Fig. 2 the CTRL and KO dissected digestive tract with the stomach, duodenum and pancreas and added a dashed line to demarcate the pancreas. A whole-mount IF staining at this stage with Pdx1 would not stain the whole pancreas, but only the beta-cells. Pdx1 expression becomes confined to the islets before birth.

Where is sox17 normally expressed compared to Pdx1 ? again they should not assume that all readers know pancreatic development.

Fig. 3c shows whole-mount IF of Sox17 in control embryos. At E10.5, Sox17 marks one of the two ventral pancreatic buds, which is negative for Pdx1, and will contribute to the gall-bladder. To help the reader, we have now added dotted lines to demarcate the buds and an asterisk to indicate complete absence of Sox17 in the Robo mutant embryo.

Figure 2a : the authors only show sections but then talk about pancreas volume. Did they image and quantified all sections ? Here again, whole-mount staining would be a real plus.

We agree with the reviewer that whole-mount IF images are sometimes more suited for volume analysis. This is why we used whole-mount IF images to measure bud volumes in Figure 3. However, since in Figure 2a we measured fluorescence intensity in single cells throughout the entire liver and pancreatic bud, we found more accurate to perform the measurement on sections and we quantified all cells in all sections. This is explained in the Method Section, page 19.

The caspase data (Fig3A) are not very convincing and they should show Caspase3 staining alone (low mag) in addition to the merge.

The single channel for the Caspase3 staining (alone at low magnification) is now shown in new Supplementary Fig. 3.

The quantification should also be by square micrometers (or cubic micrometers) and not “by area” (what is the size of the area ? the legend does not help).

We apologize for not clarifying this in our previous submission. We have now added the square micrometers (for the area) or cubic micrometers (for the volume) to all Figures.

The mouse ESC data have no interest (they could for instance have silenced Robo expression or added Slit molecules) and should be either better used (more experiments) or deleted. Adding Robo-Fc does not tell much of Slit/Robo signaling unless they show that some Slits are present in the supernatant.

We thank the reviewer for pointing this out. We now showed that Slit ligands are expressed in mES cultures differentiated into pancreatic endoderm (see new Supplementary Fig.1). This was the rationale for using a Robo2-Fc chimera in this model and explains why we cannot not add more Slit molecules to the cultures. In the revised manuscript, we further explained the biological activity of Robo2-Fc chimera in other cellular contexts, included additional citations. In addition, we provided IF analysis for Pdx1 and Nkx6.1 in mES cultures differentiated into pancreatic endoderm (see new Fig. 6 and Supplementary Fig. 6). In line with the RT-qPCR results, we found significant reduction of Pdx1-positive cells upon treatment of the differentiating cells with recombinant Robo2-Fc chimera for blocking Robo signaling.

How could Robo and YAP/TEAD be linked molecularly to Robo? No mechanism is proposed or tested. Again the data were just obtained using Robo1/2 null embryos but not with the Robo1/2PAdelta embryos.

We thank the reviewer for this important suggestion. We have included in the revised manuscript IF analyses of cellular localization of active Yap *in vivo* in the mouse ventral pancreas of Robo1/2 KO and Robo^{PaΔ} as well as in mESC exposed to Robo2-Fc. The new analysis shows reduced nuclear Yap in the absence of Robo signalling, which supports previous Tead IF and luciferase results. The data has been added in new Figure 6 and new Supplementary Figure 6.

In the new Discussion Section, we proposed a potential link between the two pathways through actin cytoskeleton.

Other points

“loxP-flanked Robo2 allele and the Pdx1-Cre” transgenic strain in a Robo1-

deficient background8, 26,” I did not find the description of this double mutant line and its validation in the two references (just on the single KO). How was it generated and what was the source of the Robo1 mutant?

We now better explained the generation of the Robo^{PaΔ} embryos in the new Material Section and inserted all relevant references in Results and Methods sections.

Is cre only expressed in the pancreas in Pdx1-Cre mice? is GFP detected outside the pancreas in Pdx1-Cre;R26RH2B-GFP embryos (this could also be genetic background dependent). Only E12.5 is shown but later stages should also be studied to confirm the specificity of Cre expression.

The Pdx1-Cre (Hingorani et al. 2003 Cancer Cell) transgenic strain is widely used in the field by many different groups. Also, the efficiency of recombination of this line has been previously published by other labs (Hingorani et al. 2003, Spence et al. 2009) and by us (Petzold et al. 2013). We show here in Fig. 5c Pdx1-Cre activity in the pancreas, portion of the duodenum and antral stomach, which perfectly overlaps with endogenous Pdx1 expression.

Additionally, we have extended the analysis of Pdx1-descendant cells to E14.5 and E18.5 and added the lineage tracing at additional time points to the new Figure 5.

Supplementary Figure 4b, c: the data are not quantified and from the pictures, the number of axons seems much higher in the mutant, and it is impossible to tell anything from the PECAM staining.

Better quality IF images have been included for the TuJ1 and Pecam staining and an additional stage has been analysed (see new Supplementary Figure 5).

“Tead levels were reduced specifically in Robo1/2 KO ventral pancreatic bud compared to controls, while the signal intensity was unchanged in dorsal pancreatic rudiments at this early stage (Fig. 6b-d).” This is basically impossible to tell from the data. Where are dorsal and ventral? Moreover, Tead expression seems to be decrease everywhere on Fig 6c.

We apologize with the reviewer if the images were misleading. We have replaced the Tead immunofluorescence images, included E-cadherin as “internal control” for the quality of the staining in the epithelium, labelled separately ventral and dorsal buds and added the quantification of the Fluorescence intensity levels in dorsal pancreas (see new Figure 7).

Why a mix of Slit1, Slit2 and Slit3 was added to the 203 cells for the luciferase assay? How was the optimal concentration determined (why 300 ng/ml? were lower doses tested?). Are the same results obtained with Robo1?

In *in vitro* experiments Slits ligands are often used together because of redundancy; the range of concentration we used is consistent with previous

data obtained with Slit proteins in different systems (see Schubert et al Int J Mol Med. 2012 Nov;30(5):1133-7; Rama et al. Nat Med. 2015 May;21(5):483-91; Zhou et al. Nature. 2013 Sep 5; 501(7465): 107–111).

Robo1 has no transcriptional activity in the Luciferase assay; this is now included in new Figure 7.

Ref 47 is incorrect : 47. Prévot Pea.???

We apologize with the reviewer, in our previous submission there were some typos in the references probably due to the automated formatting of Endnote. This has been now edited.

Reviewers' Comments:

Reviewer #1:

Remarks to the Author:

Comments on Nat Comm 162910-1

“Robo signalling pathway functions as gatekeeper of pancreatic identity”

Esco et al.

The authors responded to the comments and suggestions of 3 referees. Many of the comments are quite detailed and the authors are to be commended for attention to each and every concern. Reading through all the comments/suggestions and the author's' responses, I think the paper is improved and convincing.

General comments:

-Overall, the quality of the data is quite high: the stained histology and IF sections are nicely presented and support the conclusions the authors wish to make.

-Some parts of the previous version have been removed and I think this strengthens the paper.

Overall, the authors have attended to the 3 reviewers comments with satisfactory results. This paper nicely shows that Robo1/2 expression is important for normal pancreatic development, a a conclusion not previously demonstrated. They focus their work on “pancreatic progenitor identity” and make a convincing case that Robo and its signals in the TEAD pathway are essential for normal development.

One final point: a very recent publication (July 2018) on ROBO shows a role in endocrine cell sorting as it relates to adult pancreatic architecture. Those findings [Adams et al, Scientific Reports 8: 10876 (2018)] complement and are consistent with the present manuscript. I should like to suggest the authors consider adding this reference and a brief discussion of how those findings relate to the present work.

Reviewer #2:

Remarks to the Author:

In the revised manuscript, the authors' have addressed most if not all of the comments and concerns raised by this reviewer. The revised and newly added data are overall convincing and satisfactory, except the following few issues that still remain to be clarified.

- The authors mention that Tgif2 expression was not altered in Robo1/2 mutant pancreas and thus conclude that Robo signaling was not upstream of Tgif2 induction. They should also examine whether or not Robo signaling can be downstream of Tgif2

- Figure 1b. Even in the revised figure, it is still not clearly presented as to whether the gene expression levels at E10.5 and E14.5 can be directly compared. The authors should clearly indicate what were the “reference values” (corresponding to $y = 1$) used to calculate the Relative fold change in each of those two panels. Alternatively, the panels for E10.5 and E14.5 should be combined together to form one panel.

- Supplementary Figure 6b. The P-value for statistical comparison should be provided.

Reviewer #3:

Remarks to the Author:

The authors have done their best to address my concerns and I am satisfied with the revision.

Response to reviewer comments

We thank the reviewers for finding the revised manuscript now suitable for publication at *Nature Communications*. Below is a point-by-point response to the new reviewers' concerns with our responses shown in blue.

Reviewer #1 (Remarks to the Author):

The authors responded to the comments and suggestions of 3 referees. Many of the comments are quite detailed and the authors are to be commended for attention to each and every concern. Reading through all the comments/suggestions and the author's responses, I think the paper is improved and convincing.

General comments:

-Overall, the quality of the data is quite high: the stained histology and IF sections are nicely presented and support the conclusions the authors wish to make.

-Some parts of the previous version have been removed and I think this strengthens the paper.

Overall, the authors have attended to the 3 reviewers comments with satisfactory results. This paper nicely shows that Robo1/2 expression is important for normal pancreatic development, a conclusion not previously demonstrated. They focus their work on "pancreatic progenitor identity" and make a convincing case that Robo and its signals in the TEAD pathway are essential for normal development.

One final point: a very recent publication (July 2018) on ROBO shows a role in endocrine cell sorting as it relates to adult pancreatic architecture. Those findings [Adams et al, *Scientific Reports* 8: 10876 (2018)] complement and are consistent with the present manuscript. I should like to suggest the authors consider adding this reference and a brief discussion of how those findings relate to the present work.

We thank the reviewer for pointing this out. Adams et al, *Scientific Reports* 8: 10876 (2018) is cited in our manuscript and the findings now discussed in the revised Discussion section.

--

Reviewer #2 (Remarks to the Author):

In the revised manuscript, the authors' have addressed most if not all of the comments and concerns raised by this reviewer. The revised and newly added data are overall convincing and satisfactory, except the following few issues that still remain to be clarified.

- The authors mention that Tgif2 expression was not altered in Robo1/2 mutant pancreas and thus conclude that Robo signaling was not upstream of Tgif2 induction. They should also examine whether or not Robo signaling can be downstream of Tgif2.

In a previous transcriptome analysis (Cerdeira-Esteban et al. 2017), Robo genes did not qualify as targets of Tgif2 in the mouse. More recent RNAseq data from our laboratory is further characterizing downstream targets of both Tgif1 and Tgif2 in vivo in the mouse and human cells (Ruzittu and Spagnoli, unpublished), but we believe that this is out of the scope of this present study.

- Figure 1b. Even in the revised figure, it is still not clearly presented as to whether the gene expression levels at E10.5 and E14.5 can be directly compared. The authors should clearly indicate what were the “reference values” (corresponding to y =1) used to calculate the Relative fold change in each of those two panels. Alternatively, the panels for E10.5 and E14.5 should be combined together to form one panel.

In compliance with the reviewer’s request, we have now combined the panels for E10.5 and E14.5 expression into one panel (see new Figure 1b).

- Supplementary Figure 6b. The P-value for statistical comparison should be provided.

We have added “ns” in Supplementary Figure 6b.

--

Reviewer #3 (Remarks to the Author):

The authors have done their best to address my concerns and I am satisfied with the revision.